



# Virtual joint field campaign: a framework of synthetic landscapes to assess multiscale measurement methods of water storage

Till Francke[1], Cosimo Brogi[2], Alby Duarte Rocha[3], Michael Förster[3], Maik Heistermann[1], Markus Köhli[4], Daniel Rasche[5], Marvin Reich[5], Paul Schattan[6,7], Lena Scheiffele[1], and Martin Schrön[8]

[1]Institute of Environmental Science and Geography, University of Potsdam, Karl-Liebknecht-Straße 24–25, 14476 Potsdam, Germany
[2]Agrosphere Institute (IBG-3), Forschungszentrum Jülich GmbH, 52425 Jülich, Germany
[3]Geoinformation in Environmental Planning Lab, Technische Universität Berlin, Berlin, Germany
[4]Physikalisches Institut, Heidelberg University, Im Neuenheimer Feld 226, 69120 Heidelberg, Germany
[5]GFZ - German Research Centre for Geosciences, section Hydrology, Telegrafenberg, 14473, Potsdam, Germany
[6]Institute of Geography, University of Innsbruck, Innrain 52f, 6020, Innsbruck, Austria
[7]Institute of Hydrology and Water Management (HyWa), University of Natural Resources and Life Sciences (BOKU), Muthgasse 18, 1190, Vienna, Austria
[8]UFZ - Helmholtz Centre for Environmental Research GmbH, Dep. Monitoring and Exploration Technologies, Permoserstr. 15, 04318, Leipzig, Germany

Correspondence to: Till Francke (francke@uni-potsdam.de)

**Abstract.** The major challenge of multiscale measurement methods beyond the point scale is their complex interpretation in the light of landscape heterogeneity. For example, methods like cosmic-ray neutron sensing, remote sensing, or hydrogravimetry are all able to provide an integral value on the water storage, representative for their individual measurement volume. A rigorous assessment of their performance is often hindered by the lack of knowledge about the truth at their corresponding scale, given the high complexity and detail of natural landscapes.

In this study we suggest a synthetic virtual landscape that allows for an exact definition of all variables of interest and, consequently, constitutes the so-called "virtual truth" free of knowledge gaps. Such a landscape can be explored in various "virtual field campaigns" using "virtual sensors" that mimic the response and characteristics of actual devices. We use dedicated physically-based models to simulate the signal a sensor would receive. These model outputs termed "virtual observations" can be explored and also allow the reconstruction of water storage, which can then readily be compared to the "virtual truth". Insights from this comparison could help to better understand real measurements and their uncertainties, and to challenge accepted knowledge about signal processing and data interpretation.

The "Virtual Joint Field Campaign" is an open collaborative framework for constructing such landscapes. It comprises data and methods to create and combine different compartments of the landscape (e.g. atmosphere, soil, vegetation). The present study demonstrates virtual observations with Cosmic Ray Neutron Sensing, Hydrogravimetry, and Remote Sensing in three exemplary landscapes. It enables unprecedented opportunities for the systematic assessment of the sensor's strengths and weaknesses and even their combined use.





# 1 Introduction

Soil moisture (SM) has been acknowledged as one of the key variables governing the partition of water and energy fluxes at the soil-atmosphere interface and being a crucial link between hydrological and biogeochemical processes (Seneviratne et al., 2010; Peng et al., 2017b). Consequently, a thorough understanding and measurement of SM dynamics is desirable in a wide range of subjects, including - but not limited to - climate feedbacks, carbon respiration, agriculture, engineering, ground water recharge and flooding (Daly and Porporato, 2005; Humphrey et al., 2021; Ran et al., 2022; Scanlon et al., 2006). While numerous

measurement techniques for measuring SM at the point scale exist, each of them is affected by different specific limitations in accuracy, precision, coverage and measurement volume (Susha Lekshmi et al., 2014). Even larger uncertainties arise whenever point-based SM measurements are transferred to a larger spatial scale. Due to the high spatial variability of SM, this requires the use of a large number of sensors and the choice of their optimal locations (Corradini, 2014). Usually, electromagnetically-based sensor networks are employed for this purpose. However, hundreds of (wireless) sensors as suggested by Bogena et al. (2007)

come with considerable costs and maintenance efforts, while they are still prone to inter-sensor variability and usually not able to cover areas larger than small catchments. To overcome this lack of point-scale measurements, a number of techniques have emerged with larger spatial resolution and measurement volumes (Corradini, 2014), such as:

1. Hydrogeophysical methods: electrical resistivity tomography (ERT) (Samouëlian et al., 2005), electrical impedance spectroscopy and tomography (EIT) (Kanoun, 2018), ground penetrating radar (GPR) (Klotzsche et al., 2018; Vander-
borght et al., 2013), and electromagnetic induction (EMI) (Altdorff et al., 2018; Calamita et al., 2015; Martini et al., 2017),

2. Remote sensing (RS): passive or active microwave sensors, operated from various remote sensing platforms (Wang and Qu, 2009; Peng et al., 2017a; Mengen et al., 2021; Dorigo et al., 2017; Wigneron et al., 2017; Döpper et al., 2022b),

3. Cosmic-ray neutron sensing (CRNS): taking advantage of natural neutron fluxes and their dependence on ambient hy-
drogen pools (Zreda et al., 2008),

4. Hydrogravimetry (HG): based on its capability to observe (water) mass changes in an integrative way, gravimetry has emerged as a useful tool for hydrological observations (Creutzfeldt et al., 2010a; Kennedy et al., 2016)

.

All these methods have been developed and were tested against traditional measurements, while the above mentioned uncer-
tainties still remain. The inherent scale mismatch and spatial scale moisture variability (Famiglietti et al., 2008) vastly reduce the reliability of any performance assessments done so far (Franz et al., 2013) (reference welcome). More systematic analyses for specific conditions (e.g. the effect of varying vegetation on sensor signal) are rarely possible, as suitable datasets are still limited and can only be generated with large concerted efforts (e.g., Fersch et al., 2020; Heistermann et al., 2022b).

Thus, we are facing situations when complex and potentially powerful methods meet poor availability of ground truth data at
their corresponding scale. This dilemma is common, but not exclusive, to the Earth and environmental sciences, as detailed measurements are often unfeasible in natural systems due to large extents, inherent heterogeneity, and local intricacies. In the field





of meteorology, so-called "Observing System Simulation Experiments" (OSSEs) (Gauthier et al., 1993; Prive et al., 2021) have been an established approach for creating perfectly known systems and generating specific (but potentially error-prone) observations thereof ("virtual observations"). Examples of similar concepts can also be found for hydrological modelling (Bardossy and Singh, 2008), soil erosion assessment (Jetten et al., 1996), nutrient export (Raat et al., 2004), remote sensing (van Leeuwen et al., 2021), ecology (Fernandes et al., 2019), plant physiology (Morandage et al., 2021), hydrometry (Domeneghetti et al., 2012) and also non-Earth sciences such as epidemology (Vasiliauskaite et al., 2022).

An important aspect of these OSSEs is that they allow for free experimentation with sensor configurations to address open questions without physical limits or real investments. For example, with virtual measurements new hypotheses about the topography effect on neutron sensors could be addressed (Schattan et al., 2019) by exactly defining height, steepness, or distance of mountains; or the potential of airborne neutron sensing could be assessed without the need for constructing and navigating real-world airships (Lausch et al., 2019; Heistermann et al., 2022a).

In the context of SM moisture measurements, a suitable OSSE and its corresponding synthetic landscapes would ideally meet requirements in the following aspects:

- Scale: The extent of the synthetic landscape is larger than the support of the involved virtual sensors, i.e. it can accommodate multiple "footprints" of such measurements (e.g. cells of a remote sensing product, CRNS footprints, ...). At the same time, its spatial resolution still should allow for the representation of typical variability at the sub-footprint scale.

- Variables: The OSSE defines all variables (i.e. physical properties or landscape attributes), which are affecting the virtual observations (e.g., albedo or atmospheric transmissivity for remote sensing products).

- Usability and accessibility: The OSSE provides a large range of optional settings, so new configurations can easily be added. Existing prior data could readily be re-combined and re-analysed. All necessary scripts and data are freely available.

Compared to testing real sensors in a real environment, such an SM-related OSSE has the following advantages:

- It allows to construct landscape realizations in great numbers and variety in a consistent way. Thus, ideal conditions (which potentially cannot be found in the real world) for testing specific hypotheses are available.

- Probing the landscape realizations with the virtual sensors can be performed in an extent and density that would not be possible in real world situations. That way, we can systematically explore the impact of spatial measurement density on the success of the target variable reconstruction.

- Sensors can be tested in "best case" mode; alternatively, any amount of noise that may arise in real world situations could be imposed.

- The complete truth of the landscape realizations is known. Hence, any attempt of its reconstruction from the observations can directly be evaluated on the basis of this truth (which would, in real-world contexts, come with considerable uncertainty itself).





- The necessary steps in the OSSE setup and use involve considerable effort and/or computational resources. By standardising workflows and formats, and providing respective templates for data and scripts, studies beyond single case studies are facilitated. Preserving the respective data for later use also allows for novel multi-sensor approaches and more general multi-site analyses.

To our knowledge, such a system does not exist for SM. Therefore, this study aims to

- conceptualize a scalable framework for creating and organizing virtual landscapes,

- implement a toolkit for creating and combining compartments of such landscapes, and operating virtual sensors therein,

- present selected examples and first applications,

- propose a platform for sharing the respective tools and results.

We name this framework "Virtual Joint Field Campaign" (vJFC), underlining its purely synthetic character by mimicking observations. "Joint" refers to both the concomitant use of multiple and interdisciplinary virtual observation methods, and its open data structure for future community applications.

Section 2 describes the concept of the vJFC and its implementation. Section 3 presents three case studies tentatively analyzed with this framework.

## 2 Methods

### 2.1 Concept of vJFC framework and terminology

The general idea of the vJFC is to generate a virtual synthetic landscape from modular *compartments*, where the target properties are known (i.e. the *virtual truth*). This landscape is then probed with virtual sensors (i.e. simulation models), which provide *virtual observations*, e.g. on neutron counts, electromagnetic signatures, gravimetric signals, etc. (see Fig. 1). These virtual observations may hold an interest in their own. However, a typical workflow could also include the application of methods for reconstructing the *target variable* (specifically SM, but potentially also biomass). Comparing these to the initially defined virtual truth allows the validation of these methods.

The respective steps are explained in more detail in the following sections. At the current state, we considered three techniques for measuring SM (see section 1): Cosmic Ray Neutron Sensing (CRNS), remote sensing (RS) and hydrogravimetry (HG). This selection determines the current choice of scale and variables (see section 2.2.1). The overall idea was initially motivated from within the CRNS context, which explains its current dominance in the present case studies. However, the flexible structure of the framework allows for the integration of other techniques, too, e.g. as a part of future extensions.

### 2.2 Components of the framework

The following subsections describe the individual parts of the framework that are illustrated in Fig. 1.





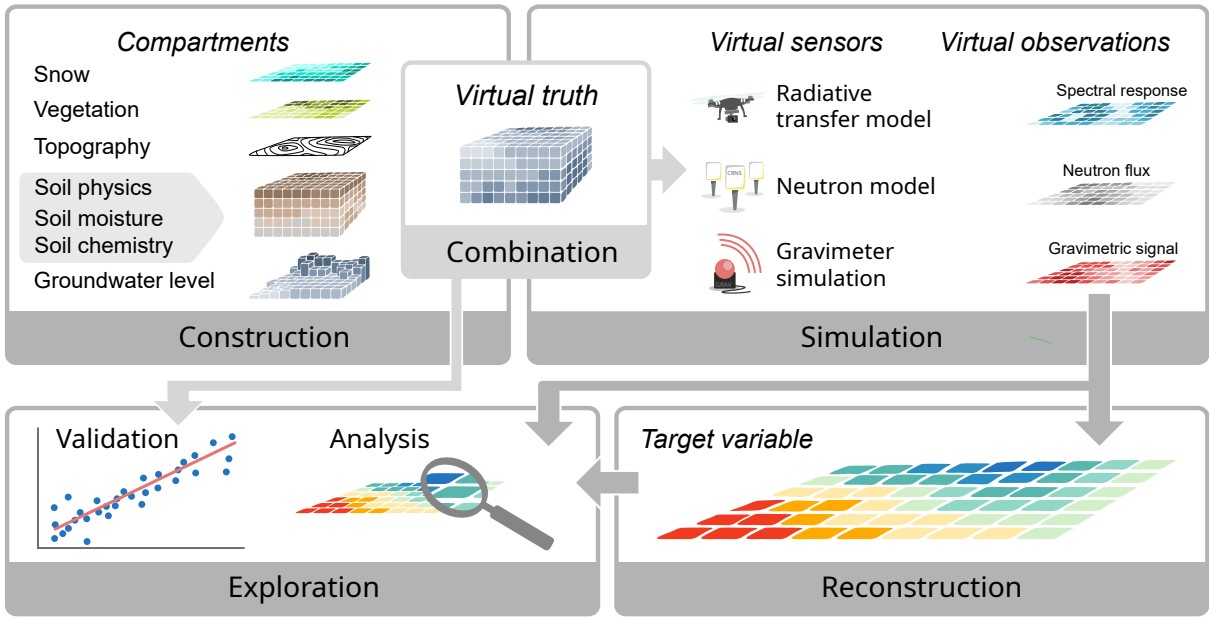

**Figure 1.** The concept of the vJFC and used terminology: "Compartments" are independent components of the landscape that can be constructed with desired characteristics. Selected compartments are merged to form a specific "realization" of a landscape. This virtual landscape can then be probed with different "virtual sensors" (i.e. models) to yield "virtual observations". These observations can drive methods for reconstructing the target variable of interest. The virtual observations directly and/or the estimates derived thereof can then be analyzed, e.g. compared to the original virtual truth.

### 2.2.1 Construction: extent, resolution, recombination

The current choice of virtual sensors (RS, CRNS, HG) requires the following minimum of physical properties to be defined:
115   elementary material composition, density and spectral characteristics. These constitute the properties that effectively determine the response of the simulated virtual sensors, see details in section 2.2.3.

The total size of the represented domain is 1000 m by 1000 m horizontally with 1 m resolution. This constitutes a compromise to reflect the typical support volume (i.e., footprint) of the sensors involved (CRNS: $10^1 \dots 10^2$ m, RS: $10^{-1} \dots 10^2$ m, HG: $10^2 \dots 10^3$ m) and the typical scale at which the natural attributes vary horizontally. Vertically, the model can be set up with no
120   fixed limit and flexible resolution to account for the fact that relevant processes may act on a wide range of vertical spatial scales, i.e. from $10^{-2}$ (e.g. penetration depth of electromagnetic waves in the top soil layers) to $10^3$ m (e.g. effects of atmospheric transmission for incoming radiation).

Due to the involved heavy computational demands, the vJFC does not consider any continuous temporal dimension, i.e. all realizations refer to a single fixed point in time. However, multiple snapshots over time could be represented with different
125   realizations.





For practical reasons in terms of modularization, the physical properties are specified via so-called compartments. These are typical "building blocks" of a landscape traditionally used in environmental sciences, e.g. *relief*, *vegetation* or *soil properties*. Depending on their role, they carry one or more of the above mentioned physical properties.

To represent the most influential landscape properties in an independent manner, the vJFC uses the compartments 'atmosphere', 'relief', 'snow', 'soil_physics', 'SM_distribution', 'groundwater' and 'vegetation' (see Table 1). Each of these compartments defines specific properties in certain parts of the virtual landscape, e.g. 'atmosphere' the material and density of the atmospheric layer.

For each compartment, multiple scenarios can exist, e.g. for 'snow' the scenarios 'none' (no snow) and 'homogeneous_0_5' (0.5 m of homogeneous snow cover). All available options are listed in the central description table (see section 4). The compartments are mostly independent of each other. However, some compartments have an implicit dependence to prevent implausible values (e.g. 'SM_distribution' depends on 'soil_physics' to ensure that soil moisture does not exceed available pore space).

Finally, some compartments may serve special purposes: 'detector' defines the arrangement of the virtual CRNS-detectors; 'pattern' is a meta-component defining spatial patterns, which can be referred to in the construction of other compartments to create coherent spatial patterns.

**Table 1.** Summary of implemented compartments serving as building blocks for assembling realizations of the virtual truth

|  | Component | number of scenarios available |
|---|---|---|
| general | atmosphere | 1 |
|  | relief | 5 |
|  | snow | 2 |
|  | soil_physics | 7 |
|  | SM_distribution | 11 |
|  | groundwater | 1 |
|  | vegetation | 5 |
| meta | detector | 5 |
|  | pattern | 3 |

The compartments constitute the elementary "building blocks" that can be combined freely to compose a realization of the virtual truth, as described in the next section.

### 2.2.2 Combination: Creating the virtual truth

Combining the compartments merges a set of these "building blocks" (e.g. relief, vegetation, snow layer) to form a specific realization of the virtual truth, represented by a 3D-datacube of the respective physical properties. Three cases of combination can be distinguished:





a) no spatial overlap: the compartments are simply combined in the joint datacube without further interaction, e.g. compartment "atmosphere" over a flat "relief".

b) stacking: an "uneven" compartment is overlaid with a second one, e.g. "soil_physics" on a hilly "relief", "snow layer" on a layer of heterogeneous "vegetation".

c) merging: a prior compartment's properties are replaced or modified by overlaying a secondary layer, e.g. voxels of "vegetation" replace voxels of "atmosphere" during their combination, "SM_distribution" alters properties of "soil_physics".

So far, 17 realizations have been generated based on the available compartments described in the previous section. Among these, the three case studies described in detail (see section 3) can be found.

### 2.2.3 Virtual observations: Probing the virtual truth with virtual sensors

The data cube containing the "virtual truth" specifies all variables that influence the response of the sensor, e.g., the spatial distribution of mass that affects a gravimeter. The virtual sensor response is computed using dedicated models. For this purpose, the respective model input files are generated from the virtual truth (e.g., 3d-matrix of mass distribution for a gravimetric model). Please note that model preprocessing might imply to map certain extra variables of the synthetic truth to a comprehensive set of parameters as required by the respective model. E.g., for the radiative transfer model, a simple vegetation type such as 'forest' needs to be mapped to a plethora of vegetation properties such as leaf area index, height, leaf pigment content, etc.

In addition to the three case studies (see section 3), some, but not all of the other realizations have yet been probed with the virtual sensors; the respective availability is listed in the central description table (see section 4).

So far, the following models serve to generate the output of the three virtual sensors.

**CRNS: Neutron transport model URANOS**

The method of Cosmic-Ray Neutron Sensing (CRNS) has been introduced by Zreda et al. (2008) and relies on the measurement of ground-albedo neutrons. The density of these cosmogenic neutrons in air is sensitive to the abundance of hydrogen in the vicinity of the neutron detector. Thus, soil moisture can be estimated within a footprint of considerable horizontal (200–500 m) and vertical (20–50 cm) extent (Schrön et al., 2017).

For generating the virtual observations, neutron transport simulations were conducted using the Monte Carlo code URANOS (Köhli et al., 2023), which was specifically designed for modeling neutron interactions within a natural environment. The standard calculation routine features a ray-casting algorithm for a single neutron propagation and a voxel engine. To simulate the physics of neutron interactions with atoms, URANOS makes use of a combination of different data bases including ENDF/B-VIII (Chadwick et al., 2011; Brown et al., 2018; Watanabe et al., 2011). Instead of extensively propagating particle showers in atmospheric cascades, URANOS uses the analytically described cosmic-ray neutron spectra from Sato (2016). This approach considerably reduces the computational effort while the effect on the accuracy of the results are negligible for most CRNS applications. To mimic the response function of real CRNS instruments, URANOS implemented virtual detector



characteristics of commonly used sensors (Köhli et al., 2018). This option also serves for representing the virtual CRNS measurements in the presented case studies and for simplicity hereafter denoted here as "epithermal counts". Concerning spatial discretisation, URANOS uses a voxel concept which transforms the simulation domain into a stack of 3D pixels, each of which

can contain materials like air, soil, rock, or snow with variable densities and porosities. This allows for directly transferring one or multiple input matrices of predefined material values to a geometry definition that can be interpreted by the model. With this concept, the model is generated with a stack of layers of variable height, resolution, and material but with common vertical and lateral extent. In a three-dimensional simulation domain, the virtual detector layers are also superimposed over voxels of air. Typical output options are neutron densities in different energy ranges from thermal to high-energy domains, entire neutron

tracks through the virtual landscape, or tracks in areas of interest. URANOS can by executed from the command line with the setup-specific configuration files to efficiently operate on desktop or high-performance architectures (e.g., Schnicke et al., 2024). A graphical user interface allows to conveniently configure the model parameters and visualise the simulations during runtime. In this study, URANOS version 1.25 was used.

**RS: Radiative transfer model**

Remote sensing exploits the fact that the presence of water causes distinct changes in the spectra of the reflected electromagnetic radiation. These changes are caused by the direct influence of the water on the surface reflectance, but also indirectly as an effect of the response of vegetation to water stress.

For the vJFC, the Soil Canopy Observation, Photochemistry and Energy Fluxes model (SCOPE) was used for simulating reflectance according to soil and vegetation properties (Yang et al., 2021; van der Tol et al., 2009). SCOPE combines seven ra-

diative transfer models to simulate spectra in the optical and thermal domains for soil and vegetation surfaces (leaf and canopy). The simulation starts by calculating soil reflectance using a Brightness Shape Moisture (BSM) model. Then, it calculates leaf reflectance, transmittance and fluorescence emission. These initial simulations combined with canopy structure parameters are the inputs to resolve the canopy reflectance (RTMo). The model assumes that vegetation canopies are homogeneous and horizontally infinite (1-D).

Our simulations were focused on soil and canopy reflectance in the optical domain (400–2400 nm). SCOPE utilises several input parameters, including soil, leaf, and canopy properties, sun–observer geometry and meteorological conditions. The most relevant inputs for this study are leaf biochemical and biophysical parameters such as leaf pigment, water, and dry matter contents. Also, canopy structural parameters such as leaf area index (LAI) and vegetation height (hc) play an essential role in the resultant spectral signal. Geometry parameters such as zenith angles and relative azimuth were kept as default based on the

sun incidence angle at noon in the summer as it may strongly affect the remote sensing signals above canopy. We used the R package rSCOPE (Duarte Rocha, 2022) to run the SCOPE 2.0 version (MATLAB codes).

The soil and vegetation reflectance parameters are based on the landscape properties of the realization *hexland_tracks* (see section 3). The bare soil spectra were simulated at the surface level ($z = 0$), while the cropland and forest spectral signals are a function of the root zone SM parameter at different depths ($z = -0.3$ and $z = -1$, respectively). The model inputs are presented

in Table A1, which shows the utilized SM-induced parameters for the different land cover types. The LAI was defined based





on the given land cover and vegetation height for each landuse class, while the dry matter was computed from the specified vegetation density. The SM-induced leaf parameters vary according to the volumetric soil moisture to simulate plant water stress. For bare soil surfaces, LAI and vegetation height were set to zero, so their spectral response is the sole function of ground reflectance. For forest hexagons, canopy properties dominate and ground reflectance is practically negligible, as the

soil background is not visible through the dense canopy.

For *hexland_tracks*, the SCOPE modelling results in 66 combinations of simulated spectral reflectance with SM-induced input parameters. The output was resampled to the 13 spectral bands of the Sentinel-2 satellite, as this is the most widely applied sensor type for this application. Furthermore, the vegetation indices Normalised Difference Vegetation Index (NDVI) and Normalised Difference Water Index (NDWI) were calculated. Additionally, an estimation of Vegetation Water Content

(VWC) based on the NDVI and a stem factor per vegetation type was performed. A set of simulated Sentinel-2 bands was mapped onto the respective hexagons, aggregated according to the band resolution (10 m for B04/B08 and 20 m for B08A/B11) and then the vegetation indices were calculated.

## HG: Hydrogravimetric model

Observing variations in Earth's gravitational acceleration due to changes in Newtonian attraction with relative terrestrial

gravimetry (TG) allows for the non-invasive estimation of total water storage changes, including the entire unsaturated soil zone, groundwater and surface water storage. As an integrative measurement, TG is sensitive to all (water) mass changes within a certain footprint around the instrument. The horizontal extent of this footprint highly depends on surrounding relief (Creutzfeldt et al., 2008) but 95 % of the signal is typically generated by mass variations within a radius of approximately $1 \, \mathrm{km}^2$ (Van Camp et al., 2017) while 85 % only depend on the first hundred meters. Gravity changes caused by local hydro-

logical mass variations can mask other geodetic signals of interest (Creutzfeldt et al., 2010b; Mikolaj et al., 2019). For these reasons, gravity time series are often corrected for (local) hydrological influences in order to study geophysical signal components of interest (Kobe et al., 2019). Approaches have been developed to quantify and remove hydrological effects (Creutzfeldt et al., 2010c; Mikolaj et al., 2015; Reich et al., 2019). In turn, gravity measurements can be used for investigating water storage variations from field to landscape-scale (Creutzfeldt et al., 2010a; Pfeffer et al., 2013; Hector et al., 2015; Güntner et al., 2017),

studying groundwater dynamics (Tanaka and Honda, 2018), hydrometerological extremes (Creutzfeldt et al., 2012; Delobbe et al., 2019) and even average forest evapotranspiration as demonstrated by Van Camp et al. (2016). However, distinguishing between different sources of observed integral gravity variations is difficult (Creutzfeldt et al., 2008, 2010b). This also applies for separating the gravity signals of water storage components such as variations of water in the upper and lower unsaturated zone of the soil and groundwater (Van Camp et al., 2017).

Forward modelling of gravity variations caused by different hydrological and soil physical properties (e.g. soil bulk density) in a virtual environment can therefore provide valuable information on how field measurements of gravity are affected by different kinds of gravity changes within the footprint. A general forward modelling procedure for calculating the gravity signal due to mass changes within a certain spatial domain is to set up a grid of components of Newtonian attraction $C$ which can then later be multiplied by spatially distributed values of mass. Leirião et al. (2009) introduced the nested grid approach,





a computationally-low method to generate such a grid, which is adapted in this study. Thus, gravity at each position depends on the magnitude and location of the mass change relative to the position of the virtual instrument. Following Leirião et al. (2009), three-dimensional grids of attraction $C_i$ were generated with the spatial extent of the virtual landscape at each of the 1 m-spaced locations of the landscape (with the virtual gravimeter always in its center). This leads to $1000 \times 1000$ attraction grids in total. Each attraction grid is then used to calculate the gravity effect of soil moisture and soil density variations for

its respective cell by multiplication with the 3D-grid of mass $M$. Summing over $M$ then yields the respective gravity for the cell. This procedure is carried out for each of the one million grids, resulting in the grid of gravity $G$. Average values of soil moisture and soil density were used as complementary information for the area outside of the domain of the virtual landscape, to account for the large footprint of the method and avoid edge effects. However, as TG aims at assessing the changes of mass (opposed to the mass itself), the difference of two gravity grids $G_1$ and $G_2$ needs to be considered.

### 255 2.2.4 Reconstruction: Estimating the target variable

The data obtained from the virtual sensors, i.e. the virtual observations, come in corresponding units, such as "neutron counts per second" for CRNS, spectral reflectance for RS, gravity for HG. As intended, these observations are proxies for the target variable of interest, i.e. SM or biomass. The procedures required to convert these proxies into estimates of the target variable vary between the sensors. Even for the same sensor, multiple approaches may exist, e.g. SM of bare soil may be derived

differently from thermal- or microwave-based indices. Thus, the chosen reconstruction methods are specific to the case study. They are not elaborated for the case studies presented in this paper, but of high relevance for envisioned follow-up studies.

### 2.2.5 Exploration: Analysis of virtual observations and reconstructions

The detailed data on the virtual observations allow in-depth analyses of sensor response per se, their resulting spatial resolution or the robustness of their signal when influenced by detrimental conditions (e.g. RS observations by hazy atmospheric con-

ditions). Thus, new insights in the characteristics of the virtual observations described in section 2.2.3 - neutron count rates, gravimetric field, spectral response - can be gained.

Similarly, any of such effects propagate to the reconstruction of the target variable (see section 2.2.4). As the target variable is completely known, a comprehensive picture of the error of these estimates can be obtained, potential influential factors can be identified, and correction methods can be tested. This will be conducted on three exemplary cases in the following sections.

To facilitate reproducibility and reusability of all parts of the vJFC, all necessary scripts and data are publicly available. See details in section 4.

## 3 Example case studies

The framework of the vJFC can accommodate an arbitrary number of landscape realizations to address a wide range of scientific questions. For illustrative purposes, we selected three example case studies that give insight into preliminary results:

*hexland_tracks* (synthetic landscape with maximised field-scale heterogeneity), *sierra_neutronica* (synthetic landscape with





high relief) and *Agia* (realistic Mediterranean landscape with orchards). The following sections describe the generation of the case studies and some example results to demonstrate the potential of the concept of the vJFC. The presented insights mainly serve exemplary purposes and are by no means exhaustive.

### 3.1  *hexland_tracks*: A landscape with maximized field-scale heterogeneity

Soil moisture is mainly governed by meteorological forcing, soil properties, and vegetation. On the scale of small catchments (i.e. 1 km$^2$), its variability is mainly the result of the variability of the latter two. The *hexland_tracks* case aims to compare all three sensor systems in a landscape that maximises landuse and soil heterogeneity at the field scale.

**Table 2.** Summary of landscape attributes implemented in the artificial landscape hexland_tracks

| Compartment | land cover | soil density | soil moisture | SM profile gradient | tracks | total |
|---|---|---|---|---|---|---|
| | bare | light | dry | homogeneous | none | |
| | agriculture | medium | 33% sat. | increasing | gravel | |
| | forest | dense | 66% sat. | decreasing | asphalt | |
| | | | saturated | | | |
| # levels | 3 | 3 | 8 | | 3 | 75 |
| | | *combinable* | | | *exclusive* | |

*hexland_tracks* includes pronounced variability of land cover, soil density, soil moisture and the vertical soil moisture gradient. Each of these compartments (or attributes thereof) is implemented in 3–4 levels, covering typical ranges of the respective
attribute (see summary in Table 2).

This variability is realized on field-scale sized plots (i.e. units of roughly 1.3 ha). These units are arranged as uniform hexagons with 72.1 m radius (Fig. 2). This pattern exploits the advantages of a hexagonal design suggested by Birch et al. (2007): a low perimeter to area ratio, less neighbours for points at the perimeter (maximum of two), and a higher number of possible combinations of adjacent polygons. Additionally, the borders are formed by three types of tracks (none, gravel,
asphalt), assigned to the track orientations SW-NE, NW-SW and N-S, respectively.

### 3.1.1  Results: Representation of spatial heterogeneity by neutron sensing, hydrogravimetry and remote sensing

The realization *hexland_tracks* has been probed by all three virtual sensors (CRNS, RS and HG), making it a favourable example for exploring the complementary properties of these methods.

The virtual CRNS observations consist of neutron count rates at different energy levels and altitudes. The shown results
assume that each voxel at a certain altitude (ground level and multiple altitudes) is a single CRNS sensor. This means, that any



**Figure 2.** The landscape realization *hexland_tracks* mimics a landscape with large variability in soil, vegetation and soil moisture conditions*
on the field scale, implemented on the scale of hexagons. It also features three levels of tracks on polygon boundaries (none, gravel, asphalt).
* ">": soil moisture decreasing with depth, "<": soil moisture increasing with depth, "=": homogeneous soil moisture.



interaction between closely-spaced CRNS sensors that would in reality appear are disregarded here. Fig. 3 shows the counts of epithermal neutrons, which is the energy level most commonly used to infer soil moisture. As mentioned before, the term epithermal refers to the neutron flux as would be measured by a CRNS instrument. The spatial pattern of these counts from the ground (i.e. at 1-2 m, Fig. 3 top) is clearly dominated by the hexagonal pattern of the prescribed hydrotopes and their soil

moisture. Despite the typical footprint radius (usually considered to range from 130 to 240 m, depending on soil moisture, see Köhli et al., 2015), the results show that the signal of each virtual sensor is dominated by its immediate surroundings. This effect is especially apparent for the 4 m-wide North-South-oriented asphalt roads and, to a lesser degree, the NW-SE-oriented gravel tracks. This local signal change has been called the "road effect" (Schrön et al., 2018) which implies a strong measurement bias when mobile ("roving") CRNS sensors move along a road (unless elaborate corrections are applied) or when

stationary sensors were placed at field borders with different landuse (Schrön et al., 2017). The vJFC dataset would allow for a more detailed assessment of this phenomenon.

Beside these near-range effects, the influence of the large CRNS footprint is evident especially at boundaries of hexagons with very contrasting soil moisture: Both high and low count rates are smoothed towards their contrasting neighbour. This smoothing effect is even more pronounced for airborne detectors operated at 30 m altitude (Fig. 3, bottom): mean count rates

decrease and smaller features such as tracks can no longer be discerned, while the general pattern of the hexagons is still well preserved. This confirms the potential of airborne CRNS (e.g., Schrön, 2017; Heistermann et al., 2022a), if the accompanying challenges such as stable flight altitude can be resolved. Additionally, land cover effects can apparently alter the CRNS signal differently, as attested by the two distinct orange-coloured adjacent hexagons NW of the centre with identical low SM and corresponding high count rates at the ground. At 30 m, however, the more northern hexagon covered by forest (cf. Fig. 2)

experiences only intermediate count rates, which also indicates that specific corrections need to be developed for such a constellation. The more extended series of layers of airborne CRNS instruments in Fig. A2 shows directly how the footprint extends with increasing height over ground. While the hexagons are clearly distinguishable for a system typically placed in 1 m distance to the soil, at 60 m elevation the contrast between the different soil moisture topologies vanishes. For the 200 m layer, the virtual detector averages over nearly half of the domain. While airborne CRNS increases the footprint, the dynamics of the

signal, however, is reduced as with increasing elevation the ground signal also gets damped. More importantly, these results suggest that the near-field effects disappear on a scale equal to the height of the instrument.

Regarding remote sensing, the modelled spectra (Fig. A1 in Annex) clearly indicate that the direct spectral signal of soil moisture is only preserved for bare soil. However, when plant water stress is considered, soil moisture-induced effects in the spectra can also be observed for cropland and forest sites.

The simulated spectral indices and the reconstructed vegetation water content show different aspects of the virtual landscape (Fig. 4). The VWC allows for a clear distinction between the three land cover classes as it is dominated by the underlying assumptions on the stem factor constant per vegetation type. It does not distinguish heterogeneous VWC, which would be expected from different SM. Among the spectral indices, NDVI can surely distinguish vegetation from bare soil, but visual separation between cropland and forest or healthy and water-stressed vegetation is not straightforward. NDWI shows a smoother

graduation with its extremes at low soil moisture content (dry) in bare soil to a saturated (wet) soil in the forest.

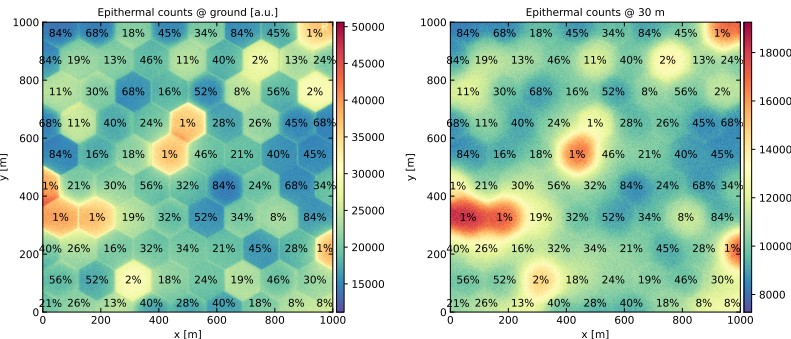

**Figure 3.** Simulated counts of epithermal neutrons near the ground (1-2 m, top) and at 30 m (bottom) in *hexland_tracks*, an artificial landscape with maximized heterogeneity. The black annotations indicate the soil moisture of the tops layer prescribed to the hexagons.

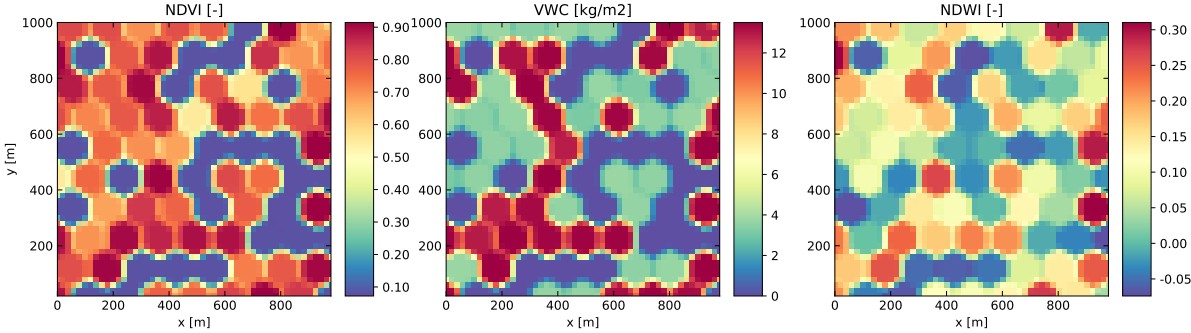

**Figure 4.** Simulated vegetation indices mimicking Sentinel 2 images for realization *hexland_tracks*. Vegetation water content (VWC, 10 m, left), Normalised Difference Vegetation Index (NDVI, 10 m, middle) and Normalised Difference Water Index (NDWI, 20 m, right).

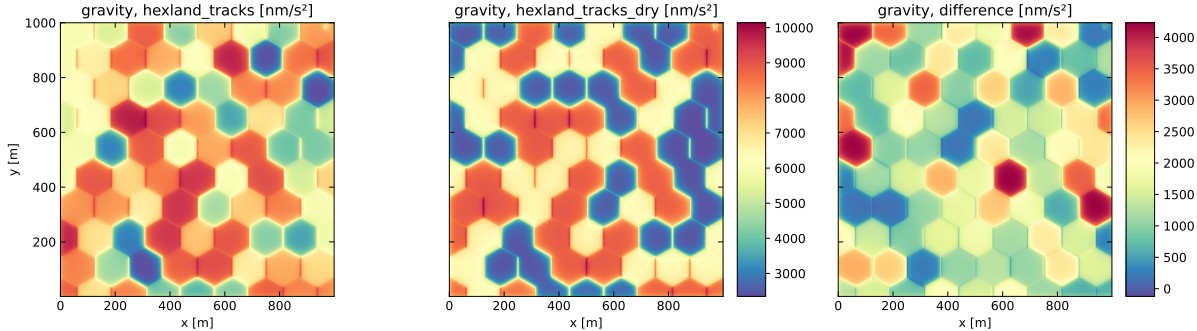

**Figure 5.** Simulated gravimetric anomalies for *Hexland_tracks* (left) and the corresponding conditions with fully depleted soil moisture storage *Hexland_tracks_dry* (center). The difference (right) allows the evaluation of static soil moisture pattern.

This confirms that the performance of remote sensing products for SM estimation cannot easily be retrieved from single indices and is strongly affected by the land cover type (Schmidt et al., 2024). Also, plant stress will be detected for different levels of soil water deficit according to the vegetation type, e.g. forest sites showing a much later spectral response induced





by soil moisture. Therefore, short-term changes in soil moisture may not be detected by RS in vegetated surfaces. Also,

soil moisture above 25% may not provoke visible changes in the spectra on vegetated surfaces Döpper et al. (2022a). The downscaling to 20 m spatial resolution creates mixed pixels, showing a border effect between different land covers and track surfaces.

For gravimetric modeling, the virtual observations represent absolute gravity values (in $nm/s^2$) resulting from the density information on all points of the virtual landscape (see Fig. 5). Modeling was carried out once for a completely dry soil with

heterogeneous density (i.e. *hexland_tracks_dry*, Fig. 5, center: dry) as well as for the same soil with the spatially varying soil moisture (i.e. *hexland_tracks*, Fig. 5, left: wet). The difference between these two grids (wet to dry) reveals gravity changes due to soil moisture information only (Fig. 5, right side). These can be attributed to changes in water content within the upper 2 m of the soil. Since the virtual landscape is flat, we can express modeled values in relation to a common assumption within hydrogravimetry: the Bouguer anomaly (Pasteka et al., 2017). This Bouguer effect describes the change in gravity due to mass

changes of a horizontally flat and infinite layer. Adding such a layer with a vertical extent of 1 m (entirely filled with water) would cause gravity to increase by 420 $nm/s^2$. The highest gravity change in the virtual observations is 320 $nm/s^2$, which is clearly below the theoretical change for the equivalent of 2 m of water. As by far not all of the soil is filled with water, this is reasonable and furthermore verifies the magnitude of changes to be expected.

88% of the observed gravity changes are above 50 $nm/s^2$, which is the typical instrumental precision to be expected of a

device feasible to carry out such mapping surveys (Scintrex, Ltd., 2017). This value corresponds to specifications stated by the manufacturer; depending on operators and post-processing, values as low as 10 $nm/s^2$ are feasible precision for relative gravity surveys.

If only mass changes within the first 50 cm of the soil are considered from the modelling, the portion of distinguishable values decreases to 17% (90%, for the more optimistic threshold of 10 $nm/s^2$). Nevertheless, in an environment with real relief

and well-selected gravity measurement locations, this portion would increase.

The spatial pattern of soil moisture-induced gravity changes is generally dominated by the hexagon structure of the input grids. This underlines the fact that the signal is almost exclusively subject to changes in the close vicinity of the observation point. Even the tracks are visible in the observed pattern. Evidently, a snapshot observation of a spatially heterogeneous distributed water and soil density pattern is not informative. Instead, the measurements yield their value when looking at the

changes of mass. For real-world gravimeters, this is usually done in the time domain, but the approach here illustrates this effect in the spatial domain: The very wet hexagons at the very Northwest and the Northern border can be clearly identified in the gravimetric pattern (cf. Fig. 2), confirming the usefulness of relative gravimetry. Edge effects of the virtual landscape could be effectively mitigated by extending the landscape spatially with average values as there is hardly a perceivable effect along the borders of the domain. Overall, the results of the hydrogravimetric approach show that modeled values could not only be

observed realistically with real field surveys but furthermore enable the identification of small-scale spatial patterns of (water) mass changes.



### 3.2 *sierra_neutronica*: Synthetic mountains to explore topographic effects

Relief exerts an important control on soil moisture patterns, mainly due to its influence on the generation and distribution of overland flow and the distance to the groundwater table. Additionally, altitudinal effects in meteorological forcing and slope-

or lithology-related patterns of soil hydraulic properties affect the SM distribution in mountainous areas. Thus, a thorough understanding of such heterogeneous soil moisture patterns is especially important but, unfortunately, coincides with numerous methodological, technical and practical challenges for SM-measurement methods. In the context of this study this comprises the following issues: for CRNS, altitude-dependent incoming neutron flux and topographic shading effects; for RS, effects of shadowing, variable incidence angles and atmospheric thickness; and for HG, potentially complex interactions of storages

above and below the sensor, and inhomogeneous lithology. All methods are also generally affected by increased difficulties in accessibility of mountaineous areas, which hinders systematic experimental assessments. Consequently, the *sierra_neutronica* case focuses at exploring complex topographic effects on neutron intensity at the ground.

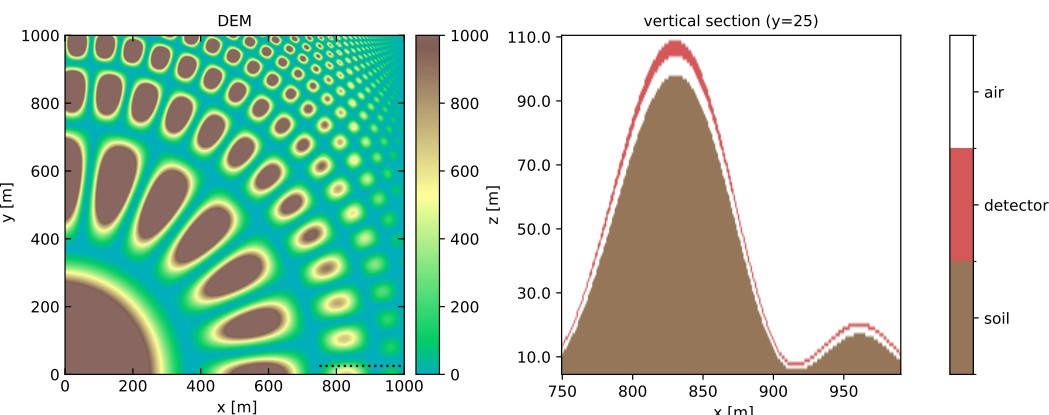

**Figure 6.** The landscape realization *sierra_neutronica* combines two gradients of various amplitudes (West-East) and frequencies (radial and tangential) to create a large variety of topographical settings. Apart from a barometric pressure gradient with altitude, all other properties are homogeneous, i.e. soil with 5.5% SM and no vegetation. The right panel shows a vertical cross section (dotted line in left panel) to illustrate the detector layer used in the neutron simulation (ignored otherwise).

For the sake of simplicity, *sierra_neutronica* features homogeneous bare soil of constant thickness and soil moisture (1000 m deep, 5.5% volumetric water content). The relief spans an altitude range of 1000 m, representing an alpine valley. It combines

two gradients of various amplitudes (West-East) and frequencies (radial and tangential), so a wide range of topographical conditions is included (see Fig. 6).

The simulation of the neutron flux in steep terrain required adjustments compared to those in flat conditions: As illustrated in Fig. 6, the detector layer was created parallel to the ground while avoiding neutron-sensitive voxels in direct or diagonal contact with the terrain surface. This resulted in multiple detector voxels per vertical columns for the steeper parts of this

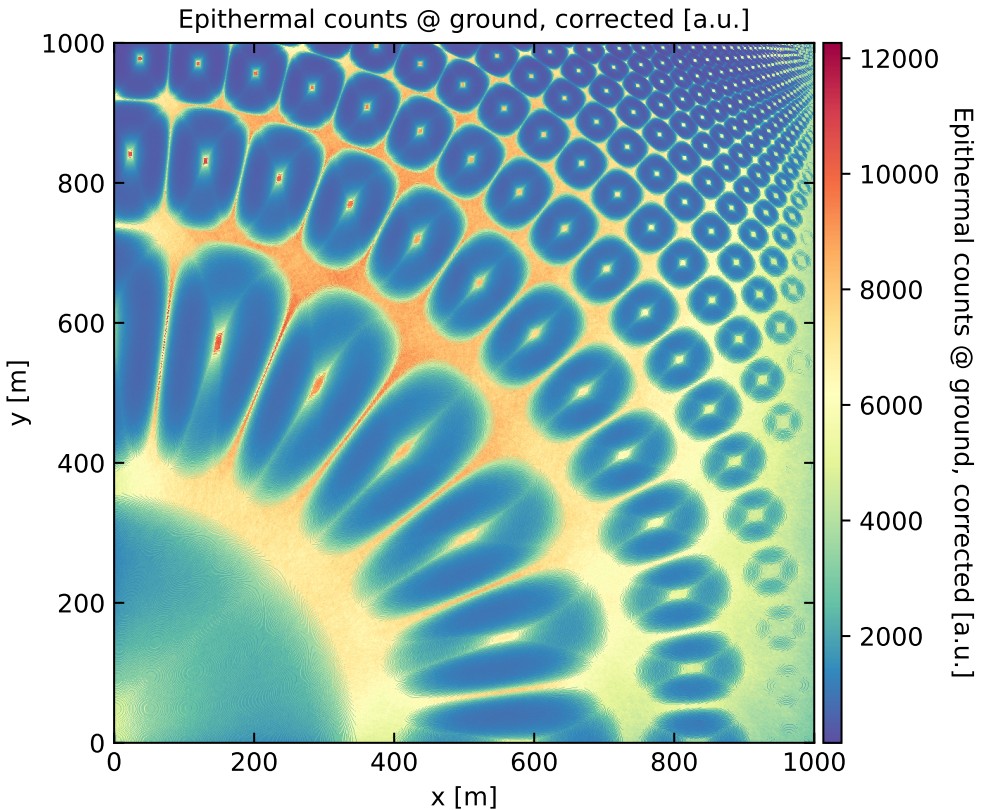

**Figure 7.** Simulated counts of epithermal neutrons in *sierra_neutronica*, an artificial mountainscape.

landscape, visible for the left hill depicted in Fig. 6. Consequently, this horizontally-varying number of detector voxels had to be accounted for by normalizing the modelled count rates accordingly.

### 3.2.1    Results: Topographic effects on CRNS

Fig. 7 shows the count rates corrected for the varying number of stacked detector voxels. For large topographic structures (i.e. mountain in the SW), the count rates show a gradual increase with altitude, which corresponds to the expected increase

in count rates caused by barometric effects. However, pronounced deviations from this trend can be observed for the more complex parts of the landscape: the wider valleys, hill tops and exposed ridges show higher count rates. Fig. 8 illustrates this effect, by showing how much (in relative terms) the modelled count rates deviate from the expected increase solely explained by simple barometric correction (Zreda et al., 2012): Apparently, deeper valleys, hilltops, and ridges experience additional effects of topographic exposure and shielding (e.g. Dunne et al., 1999; Balco, 2014; Schattan et al., 2019), which are not

directly related to pressure or altitude and deserve further detailed examination. Apparently, steep hillslopes show a lower overall neutron flux than expected from flat terrain whereas steep valleys would show an increase in count rate.

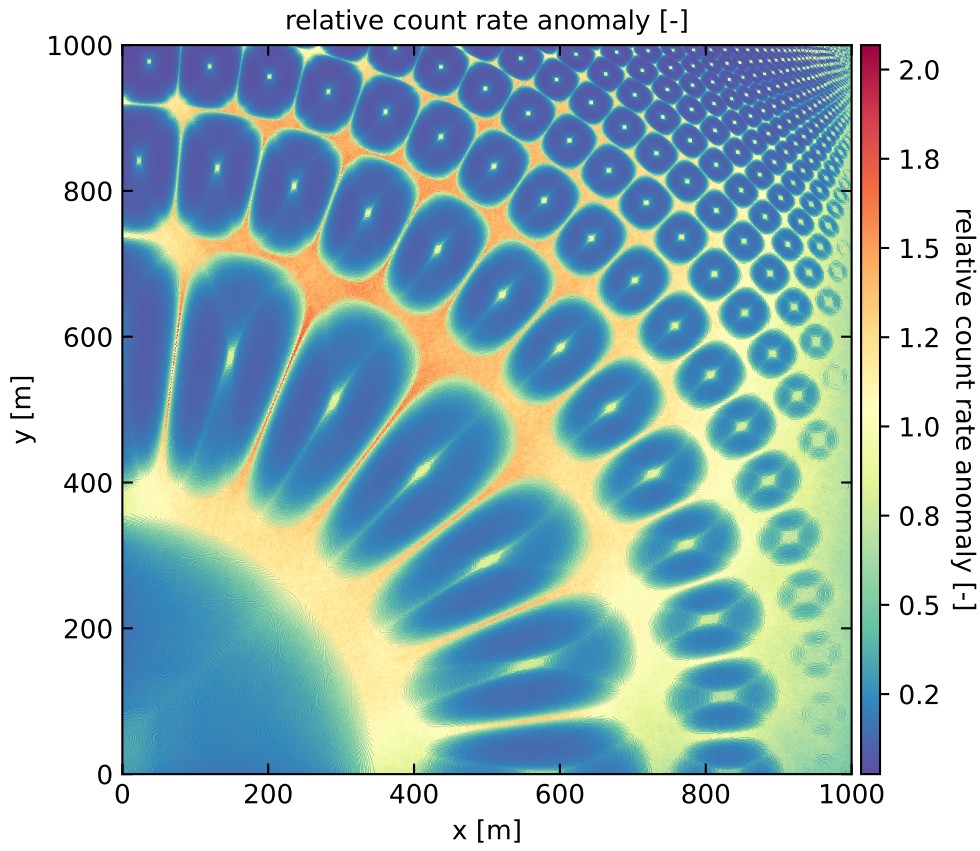

**Figure 8.** Relative anomaly in simulated counts of epithermal neutrons in *sierra_neutronica*, compared to expected pressure-corrected counts.

### 3.3 *agia*: A realistic irrigated agricultural landscape in Greece

CRNS has great potential to monitor and inform irrigation in agriculture (Franz et al., 2020; Ragab et al., 2017; Finkenbiner et al., 2019). However, it has been shown that the quality of the information provided by the CRNS strongly depends on the type

of sensor that is employed and on environmental factors such as the amount of irrigated water, the dimension of the irrigated field, and the SM conditions (Li et al., 2019; Brogi et al., 2022). The *agia* case aims to investigate how well CRNS could quantify soil moisture on irrigated plots that are embedded on a heterogeneous and rather dry environment by reproducing an agricultural setting found near the village of Agia (39.718°N, 22.741°E), (Greece). This area is characterized by a highly heterogeneous landscape, with small fields that are irrigated at different times and complex spatial variations in SM. Such an

environment is challenging for CRNS applications but can provide precious insights on sub-footprint heterogeneity and on the contribution of different land patches to the CRNS signal (Schrön et al., 2023; Brogi et al., 2022).

Within this area, in 2020, two apple orchards of approximately 1.2 ha and separated by approximately 300 m were equipped with extensive instrumentation to test the use of CRNS in irrigation management (Brogi et al., 2023). A $1 \times 1 \, \mathrm{km}^2$ domain



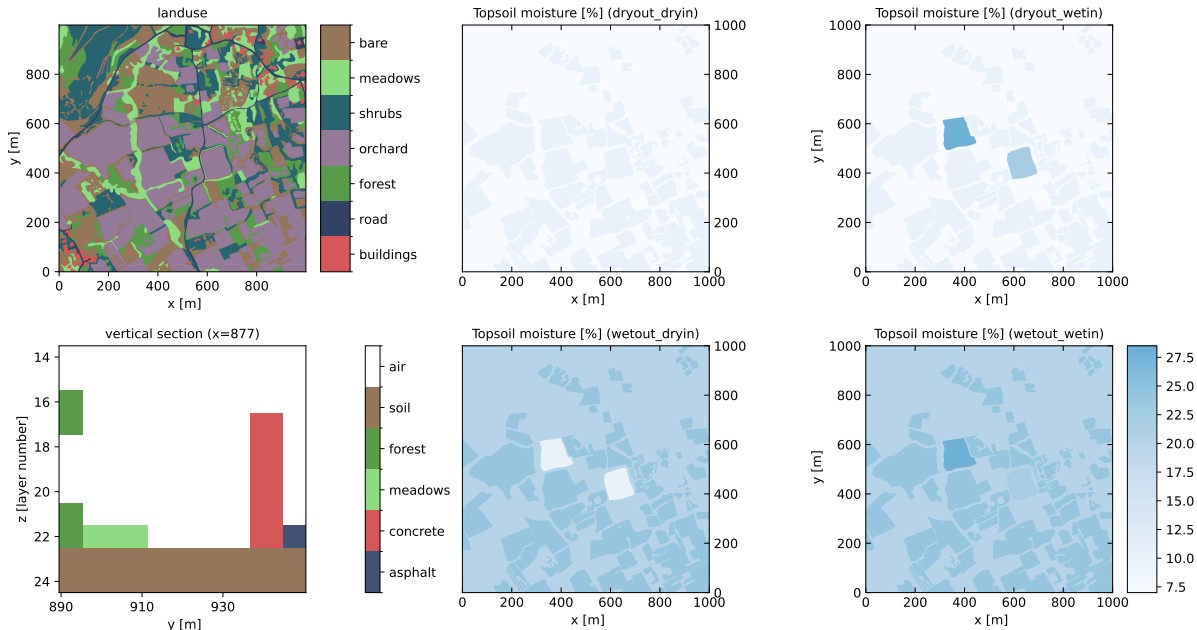

**Figure 9.** The landscape realization *agia* represents a realistic setting for a Mediterranean agricultural landscape. This figure shows the common landuse and the four soil moisture realizations, in which the surroundings and the pilot fields are set to low or high soil moisture, respectively.

(see Fig. 9) was centred between the two fields and eight land cover classes were digitised from satellite images (ESRI, 2022).
Land cover height was set to 6.0 m for trees, 4.0 m for apple orchards and buildings, 0.6 m for bushes, 0.1 m for grass and asphalt roads, and 0.0 m for bare soil and tracks. The vegetation biomass was set equal to the URANOS material code *tree gas* (3.0 g cm$^{-3}$) for apple orchards and trees, and equal to *plant gas* (5.0 g cm$^{-3}$) for bushes and grass. The soil compartment is represented by three layers of 0.3, 0.3 and 1.0 m thickness (maximum depth of 1.6 m). Soil density is horizontally homogeneous, increasing with depth (1.5, 1.6 and 1.7 g cm$^{-3}$, resp.). In the pilot irrigated fields, bulk density, porosity and other soil hydraulic
properties were obtained from soil sampling, laboratory analysis, and the use of pedotransfer functions (Rawls and Brakensiek, 1985). These helped to estimate SM at field capacity (maximum simulated SM) and wilting point (minimum simulated SM) by using the Mualem–van-Genuchten model (Van Genuchten, 1980). The surrounding irrigated fields were assumed to have SM values that are the average of the two irrigated pilot fields. In the non-irrigated areas, maximum and minimum SM were assumed to be those recorded by SM sensors positioned at 5, 20, and 50 cm depth at a single location.
Four different simulation scenarios are available. In these, the vegetation remains constant while the SM distribution varies. The scenarios are a) dry irrigated pilot fields and surrounding (*agia_dryout_dryin*), b) dry pilot fields and wet surroundings (*agia_wetout_dryin*), c) wet pilot fields and dry surroundings *agia_dryout_wetin*, and d) wet pilot fields and surroundings (*agia_wetout_wetin*). The reader is refered to Brogi et al. (2023) for a more in-depth description of the study site and the process behind the production of the data.



**3.3.1   Results: Detectability of irrigation in relation to surrounding landscape variability**

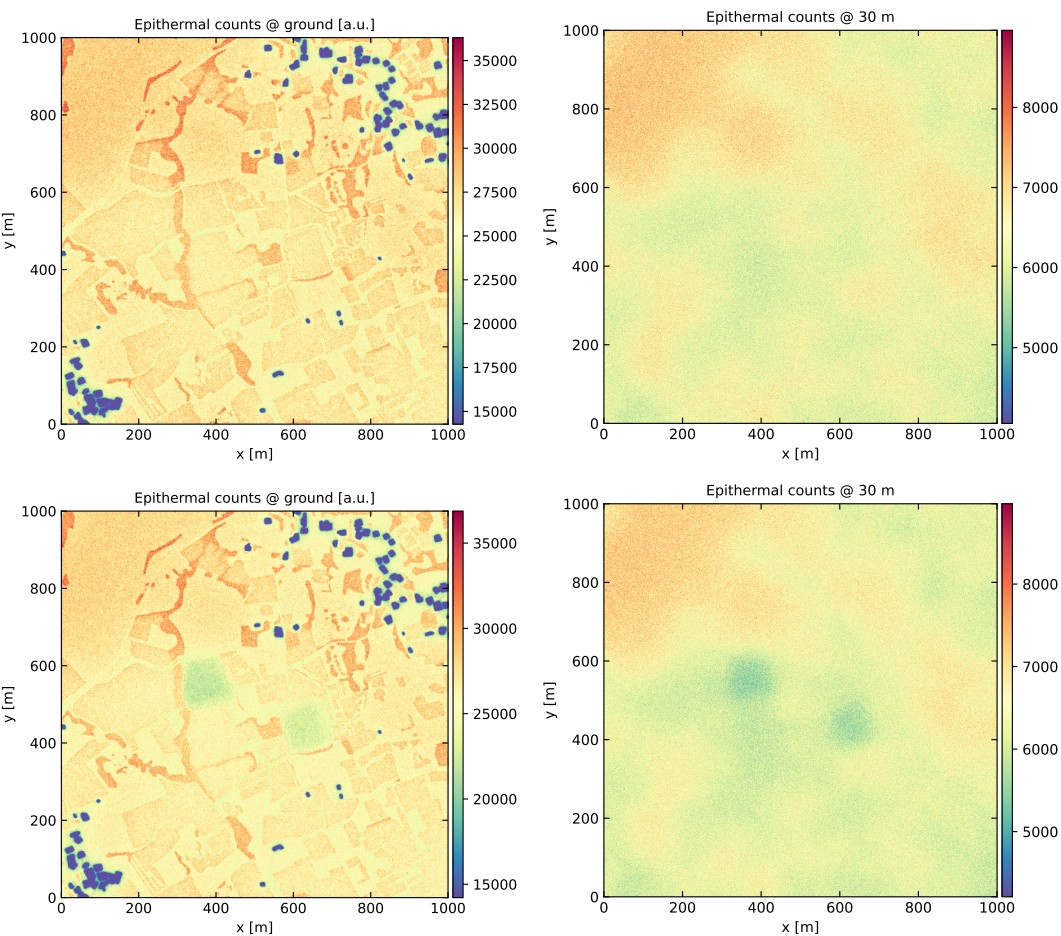

**Figure 10.** Simulated counts of epithermal neutrons near the ground (1-2 m, left) and at 30 m (right) in a realistic Mediterranean landscape under conditions of low soil moisture variability (*agia_dryout_dryin*, top row) and with irrigated fields in dry surroundings (*agia_dryout_wetin*, bottom row).

For the Agia site, four different realizations have been generated from four different scenarios of soil moisture patterns. Two of these, *agia_dryout_dryin* and *agia_dryout_wetin*, are shown in Fig. 10. The first scenario (top row) represents the generally dry SM conditions in which the investigated area can be found during the summer period while the second scenario (bottom row) shows an increased SM in the two pilot apple orchards at the center of the domain. This higher SM is the typical consequence of weekly mini-sprinkler irrigation events that is applied to the entire field and can reach up to 43 mm day$^{-1}$ in the Western pilot field and 54 mm day$^{-1}$ in the Eastern pilot field. In the simulated case, the irrigation amount in the Western pilot field is smaller than in the East one, but SM after irrigation is higher due to differences in soil properties.





The result of the SM increase in *agia_dryout_wetin* is a lower simulated count of epithermal neutrons near the ground (Fig. 10, bottom left) within the irrigated fields. This has an impact on the signal recorded by a CRNS positioned within the

irrigated field. However, the CRNS footprint is larger then the investigated 1.2 ha fields, and neutrons that have soil contact outside the field can strongly affect CRNS measurements, thus reducing or masking the effects of irrigation, especially at the borders of the field. A correction developed by Brogi et al. (2023), which uses the simulated contributions to the CRNS signal of the irrigated field and of its surroundings, can be applied to exclude the effects of the non-irrigated surroundings. The simulated realizations can provide detailed information on such contributions for the Agia site, and could be further adapted to

any other area of interest via additional realizations.

Based on these results, the contributions to the neutron count of the irrigated field and of the surroundings, which is necessary for irrigation corrections (Brogi et al., 2023), can be extracted for a CRNS placed in any position within the target fields. This also allows to simultaneously investigate multiple fields and identify a) the most suitable locations within each field, b) the minimum number of CRNS required to adequately monitor irrigation in one field or in a given agricultural area, and c) the most

suitable CRNS design in terms of response function and sensitivity. Such results have ramifications for instrument selection and for the estimation of costs and benefits that are tailored to specific areas and irrigation techniques.

The left panels of Fig. 10 show strongly heterogeneous patterns in epithermal neutron intensities for the ground-based sensor (1-2 m height) that are the effect of both SM and vegetation distribution. The right panels of Fig. 10 show the epithermal neutron counts at 30 m above the surface. As in the *hexland_tracks* case study (section 3.1.1), these patterns are much smoother

than those recorded at the ground. However, the effect of irrigation in the two pilot apple orchards is still clearly noticeable given the large drop in epithermal neutron intensities 30 m above ground. This has implications for airborne CRNS roving applications that are similar to those discussed in the case of the *hexland_tracks* realization. A general influence of the landuse, when large areas share similar cover, can be identified at this height. First, the more natural landscape of the north-west part of the domain, consisting of relatively low shrubs and grass, results in a higher density of epithermal neutrons at 30 m height.

In contrast, the agricultural areas that are principally made of orchards in the center and south-eastern parts of the domain result in a generally low epithermal density at 30 m altitude. Interestingly, the presence of buildings also results in relatively low epithermal densities at 30 m and, more strongly, at 1-2 m height within a few metres around the buildings. The patterns of count rates registered by the ground-based sensors shows their highest count rates for the meadow areas (e.g. crescent-shaped north-south structure). The count rates seem especially increased towards the borders of the meadows. Thin, more linear

shapes seem especially affected (e.g. thin meadow are in NW-sector, meadow strip south of Eastern pilot plot). It remains to be analyzed if these patterns are caused by exposure effects from surrounding higher land cover classes (similar to the phenomena observed in *sierra_neutronica*, see section 3.2.1), or they need to be attributed to interactions with the neighbouring plots.

Despite the relatively high resolution of the domain of 1 m, there are simplifications that can have an impact on the results and on the considerations made so far. Landuse voxels in the current resolution of $1 \times 1$ m dimension, and simplify certain

structures. For example, a building is represented by an homogeneous gas and this may have a different impact on the result compared to a detailed structure with separated walls and, especially, roof materials. Similarly, a tree is also represented by an homogeneous gas and does not distinguish between different organs. Although it can be expected that a more detailed and





complex representation will have an effect on the results, it has to be noted that the presented complexity is higher that that of most studies found in literature, especially given the extent of the domain and the purpose of the vJFC setup. Nonetheless,
these considerations further motivates the representation and study of realistic scenarios in an environment such as the vJFC as key insights can be gained for several CRNS applications and beyond.

## 4    Conclusions and Outlook

The "Virtual Joint Field Campaign" represents a framework to design virtual landscapes in which we can deploy virtual sensors (here CRNS, remote sensing and hydrogravimetry). Such a virtual sensor allows us to simulate the measurement of variables
(e.g. neutron intensity, reflected spectra, gravimetric anomalies) that have a well-defined relationship to an actual target variable of interest (e.g. soil moisture or biomass). Based on such virtual observation, we can explore potentials and restrictions for the spatio-temporal retrieval of such target variables at the scale of $1\,\mathrm{km}^2$, in the presence of different levels of landscape complexity. It represents, to our knowledge, the first effort to define a comprehensive "Observing System Simulation Experiments" in the context of soil moisture and biomass observations.

The presented definitions and conventions allow the reproducible use of existing setups and recombinable creation of new case studies from new or existing components. The Open Data approach, comprising the free availability of all generated data and respective scripts, provides a low barrier for the scientific community, especially when aiming at comparing or combining different sensors.

This paper presented three case studies with different thematic focus, but is not intended to analyse their outcomes in detail.
Multiple aspects which merit further analysis have been identified and outlined in the respective sections or can be perceived, e.g.:

- How good can we estimate soil moisture based on * perfect single-point calibrations * (im)perfect knowledge of biomass inventories, bulk density information?

- How good can we estimate biomass?

- How many CRNS-probes are needed for getting robust SM information at the landscape-scale? Can we give recommendations for minimum instrument sensitivity and minimum distances of CRNS sensor locations to adjacent physiographic units to obtain unbiased count rates?

- Does the reconstruction of SM (e.g. using inversion methods) based on hydrological units work? Under which premises?

- How do roving sensor applications perform in different parts of the landscape? How does the road effect (Schrön et al.,
2018) affect estimates of adjacent areas? How many and which routing of roving tracks would be needed to capture the landscape-average SM distribution? What auxiliary information is required and can be obtained alongside (e.g. use of counts of thermal neutrons for biomass estimation)?





- How do relief and vertical structure influence count rates? Are the current corrections (altitude via pressure, (exposure?)) sufficient for our purpose?

- How can a gravity survey be planned with realistic field work times (not too many measurement sites) in order to meet the requirements of capturing spatial (possibly isolated) features relevant for the area of interest?

- What could be the basis of selecting such gravity measurement sites in terms of quantity and spatial representativeness in combination with landscape properties such as topography, landuse, etc.?

- Is it feasible to capture the overall mass change dynamics in a representative way with a more precise but spatially fixed,
permanent gravity observation site (given the fact that the installed gravimeter is more precise than field units) and how valuable would this information be in combination with the other methods?

- For both, CRNS and Microwave remote sensing, SM inversely affects penetration depth, and vegetation density and structure affect the sensor signal. How could the consideration of passive and active microwave responses of different bands (X, C, S, L, P) improve the understanding of both sensors?

The presented case studies assumed perfect measurement conditions, i.e. no instrument error e.g. due to atmospheric conditions. However, as this kind of noise often poses severe restrictions on the usability of the sensor signal (e.g. due to short counting intervals for CRNS, atmospheric transmissivity for RS, instrument noise for HG), systematically assessing the limits of applicability could be greatly facilitated with the vJFC.

Additionally, the potential of multi-sensor use has hardly been investigated: (How) would RS-derived SM-estimates profit
from CRNS-derived training locations? Could the robustness of CRNS-signals be increased with additional information from RS?

The neutron simulations, in particular, required an unprecedented total computational effort, as several other realization were generated besides the presented case studies. This wealth of simulated constellations may provide valuable starting points for exploring the options to find a forward operator with low computational requirements, e.g. by training machine learning
methods on the available data.

Whatever the focus, we invite the reuse of the available data and happily accept contributions for further compartments, realizations and simulations to increase the joint value of the data set.

*Code and data availability.* To facilitate reproducibility and reusability of all parts of the vJFC (data and scripts) are publicly available for the point of writing this paper from https://doi.org/10.23728/b2share.854eda607a1d4aca8333afc24cfe593a .
As we envision a further growth of included data and methods, further amendments are constantly added to the more flexible repository at https://b2drop.eudat.eu/s/DoFfxQx6cWFSAPq. Potential users are invited to use, modify and analyse the included data and scripts. We also welcome any substantial contribution in terms of compartments, realizations and related virtual observations. By enriching the pool of available data, even more comprehensive analyses are enabled.





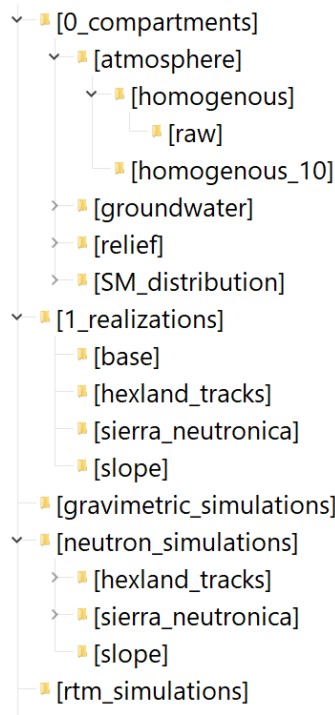

**Figure 11.** Simplified view of the folder structure holding the vJFC data

*Data*

This repository uses a hierarchical folder structure corresponding to the described compartments, realization and virtual observations (see section 2.2.1 - 2.2.3 (see Fig. 11):

- The folder "0_compartments" holds subfolders of each compartment (e.g. "atmosphere", "groundwater", etc.). Each of these sub-folders contains the different scenarios for the respective compartment (e.g. "homogeneous" representing a homogeneous atmosphere).

- The folder "1_realizations" holds the compiled realizations of the virtual landscape, e.g. *hexland_tracks* as described in section 3.

- The folders "gravimetric_simulations", "neutron_simulations" and "rtm_simulations" contain subfolders for each realization of the virtual landscape (e.g. *hexland_tracks*) as described in section 3. Each of these subfolders contains the specific model input files to simulate the virtual sensor, and the respective model outputs.

At each folder level, readme-files describe the respective entities in the folder, and other meta-data.

The top-level spreadsheet file "overview_vJFC.xlsx" provides central description tables as an overview of all available compartments,

realizations, and the status of their simulation. Data are stored in NetCDF format, as this standard is very suitable for large multidimensional data sets and allows for easy interchange with numerous other software. Details on the employed data structure in the NetCDF are described in the top-level readme file.



*Scripts and external model code*

Accordingly, the scripts for the generation of the data are organized at the same three levels evident in the folder structure:

– "compartment" scenarios are generated using scripts and data located in the respective subfolder "raw". These scripts (mostly in R-language, (R Core Team, 2018)) can be used as templates for creating other scenarios of the compartments.

    – "realizations" are compiled from the compartments using the Python package YULIA, v1.01. Besides its original functionality of systematically creating numerous sets of URANOS parameterisations, YULIA allows merging selected compartments into a realization following vJFC conventions. YULIA is available from https://gitlab.com/crns4snow/yulia.

– Model input files for simulating the "virtual observations" are generated with dedicated scripts for each virtual sensor. The resulting files reside in the folders with suffix "*_simulations":

    – The Python package YULIA (see previous section) serves for creating the files for neutron modelling with URANOS, stored in "neutron_simulations". URANOS, the model used for simulating neutron flux (see section 2.2.3), is available from https://gitlab.com/mkoehli/uranos.

    – As basis for the "gravimetric_simulations", gravity grids are generated with scripts located in the subfolder "Scripts" of the
respective realization. These Python scripts build on the Python package "hygra" (provided as zip-file in the same directory), which is also used to carry out the actual gravimetric modeling.

    – For RTM-modelling stored, the scripts can be found under the folder named "rtm_simulations/Scripts". The script is written in R (R Core Team, 2018), executing the RTM model from SCOPE (Yang et al., 2021; van der Tol et al., 2009) by running the respective MATLAB code through the R package rSCOPE (see https://github.com/AlbyDR/rSCOPE and https://github.com/
Christiaanvandertol/SCOPE/tree/master). The spectral resampling to Sentinel 2 and the vegetation index calculation were performed using the hsdar R package (e.g., Lehnert et al., 2019). The simulated NDVI, NDWI, and VWC indices for the hexland_tracks are in the same above-cited folder.

The figures generated from results of the case were generated with the scripts in "misc_scripts/illustrations_paper", which largely depend on Corny (https://git.ufz.de/CRNS/cornish_pasdy) and uranostools (https://github.com/cosmic-sense/uranostools).



# Appendix A

## A1  Simulated spectra for realization "hexland_tracks"

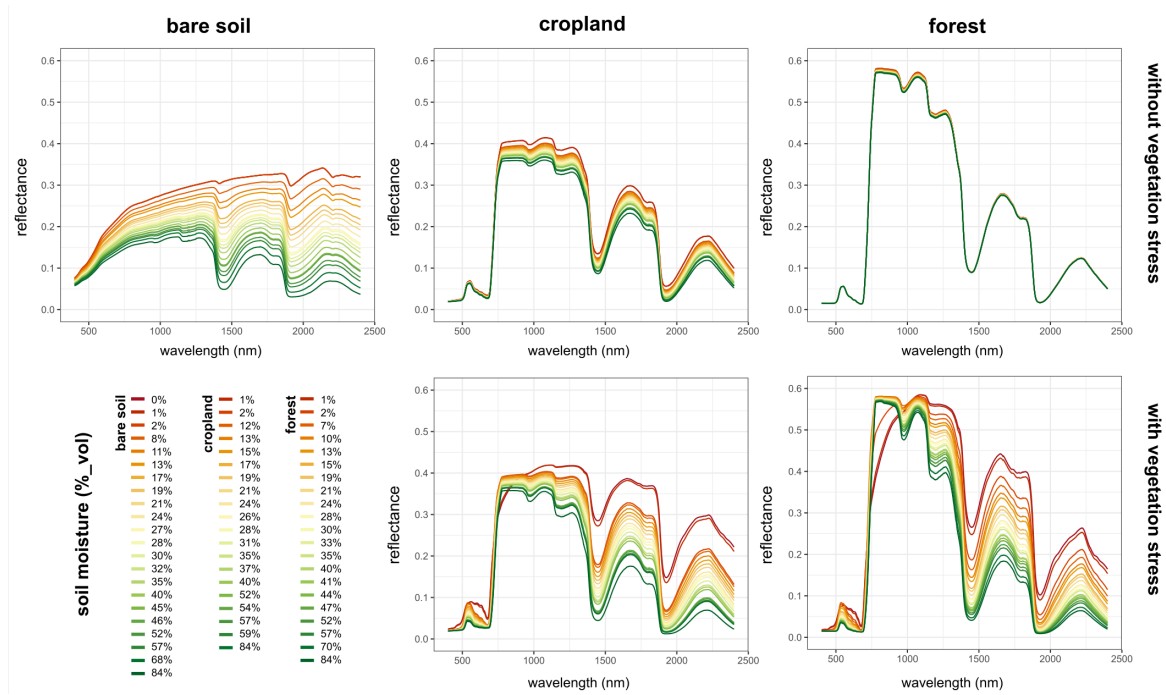

**Figure A1.** Simulated spectra of the three main landcover classes "bare soil" (top left), "cropland" (centre) and "forest" (right) in the *hexland_tracks* realization. The top row does not consider SM-related water stress in the vegetation, while the bottom row includes this effect.

*Author contributions.* TF coordinated the study, coded the major scripts for creating and merging the compartments, prepared the first draft of the manuscript and the data repository. CB prepared all necessary data for the Agia example. ADR and MF set up the RTM-model and conducted the RS simulations. TF and MH developed the basic concept of the vJFC; MH and MS supported in programming of scripts and generation of figures. MK implemented essential extensions to the neutron model, guided scientific decisions in the respective parameterizations and conducted the simulations. MK and MS provided computational resources. MR and DR designed and conducted the HG simulations. PS provided the code of YULIA as a basis for the core scripts of the vJFC. LS pushed the development of the concept in its early stages. All authors contributed to the development of the vJFC-framework and edited the final manuscript.

*Competing interests.* M. Köhli holds a CEO position at StyX Neutronica GmbH. All other authors declare no competing interests.





**Table A1.** Model input values to simulate spectra according to the land cover and vegetation water stress induced by soil moisture depletion.

| Biochemical and biophysical parameters (canopy and leaf traits) | | Land Cover | | |
| --- | --- | --- | --- | --- |
| | | Bare soil | Cropland | Forest |
| | Leaf Area Index (LAI, $m^2/m^2$) | 0* | 2 | 6 |
| LULC | Vegetation height (hc, m) | 0* | 1 | 20 |
| | Dry matter content (Cdm, $g/cm^2$) | 0 | 0.0015 | 0.0022 |
| SM | Root Zone Soil Moisture (SMC, $\%_{Vol}$)** | 0 to 0.84 | 0.01 to 0.84 | 0.01 to 0.84 |
| | Chlorophyll AB content (Cab, $\mu g/cm^2$) | 0 | 11.6 to 72.1 | 11.6 to 72.1 |
| SM-induced | Carotenoid content (Cca, $\mu g/cm^2$) | 0 | 2.3 to 14.4 | 2.3 to 14.4 |
| | Leaf water equivalent layer (Cw, cm) | 0 | 0.002 to 0.019 | 0.002 to 0.019 |
| | Senescent material fraction (Cs, 0-1 index) | 0 | 0 to 0.45 | 0 to 0.45 |

note: * 0.001 was used instead of 0 (zero) to allow running the SCOPE model. ** SM depth $z = 0$ for bare soil, $z = -0.3$ for cropland and $z = -1$ for forest.

*Acknowledgements.* This research was funded by the Deutsche Forschungsgemeinschaft (DFG, German Research Foundation), research unit FOR 2694 "Cosmic Sense", project number 357874777. The authors gratefully acknowledge Lukas Strebel's help. The authors thank Steffen Zacharias (UFZ) for helpful discussions during the preparation of this study. We would also like to thank the administration and support staff of the HPC cluster "EVE" at UFZ, in particular Ben Langenberg and Toni Harzendorf.



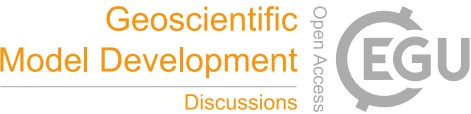

**Figure A2.** Epithermal counts measured for *hexland_tracks* realization at various sensor altitudes.

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
