# Peer review of "Virtual joint field campaign: a framework of synthetic landscapes to assess multiscale measurement methods of water storage"

_Geoscientific Model Development, 2024_

## Author Comment (AC1)

**Author Response to Editor**

**Virtual joint field campaign: a framework of synthetic landscapes to assess multiscale measurement methods of water storage**

Till Francke et al.
*Geoscientific Model Development Discuss.,* `doi:10.5194/gmd-2024-106`

RC: *Referee Comment*,    AR: *Author Response*,    ☐ Manuscript text

Dear referee,

thank you very much for your positive response, and for the time and effort spent to examine the manuscript and the data set.

Your comments are very helpful in improving the quality of our manuscript. Please find a point-by-point reply below on how we intend to implement them. If you feel that some of these changes would not satisfy the needs you indicated, we would appreciate further advice on these matters.

Kind regards,
Till Francke (on behalf of the author team)

**Comments and responses**

RC: *The paper is well written, well-structured and clear. The topic is surely of interest for the readers of Geoscientific Model Development (GMD) as the paper introduces a new virtual framework to assess measurement methods of soil moisture and biomass in controlled experiments.*

AR: We appreciate this positive perception our study.

RC: *MAJOR COMMENT 1: The virtual framework is, in principle, suitable for any measurement technique. The paper analyses three methods: cosmic-ray, remote sensing and gravimetric measurements. However, while cosmic-ray are well developed and described, this is not the case for remote sensing and gravimetric measurements. Remote sensing and gravimetric measurements are only used in the hexland_tracks experiment, cosmic-ray in all the experiments.*

AR: We agree that the framework could be suitable for various measurement techniques. We tried to demonstrate this with our case study "hexland_tracks", in which CRNS, passive optical remote sensing and hydrogravimetry are used. These examples are by no means exhaustive: active microwave remote sensing, GNSS reflectometry or ground penetrating radar are other possible examples. However, as these other methods require specific expertise, we must confine ourselves to our area of knowledge. Thus, we can only invite the respective

communities to this endeavour, but cannot cover all possible examples for the framework.

Our focus was on the presentation of the framework, not on specific case studies. We used one realization (hexland_tracks) to demonstrate the applicability *across different sensor types*. As such, the other two case studies merely underline the versatility of the framework *across different landscape realizations*, specifically in representing terrain or resembling non-artificial landscapes.

We will clarify our focus on the framework and the illustrative and selective nature of our case studies and support this with the following table:

Table 1: Overview realizations

| realization | description | generated virtual observations |
|---|---|---|
| *hexland_tracks* | synthetic landscape with max. contrasts | CRNS, remote sensing, hydrogravimetry |
| *sierra_neutronica* | synthetic landscape with high-relief | CRNS |
| *agia* | realistic Mediterranean landscape | CRNS |

**RC:** *In addition, optical remote sensing data are considered, but currently no soil moisture products from optical data are available, only results from scientific papers.*

AR: The output of the radiative transfer model is spectral reflectance (resampled to the 13 bands of Sentinel 2). It is correct that we do not use the reflectance data to retrieve a soil moisture product, but just illustrate the use of thespectral response by more common indices related to the presence of water and chlorophyll (NDV, NDWI, VWC). The same essentially applies to the virtual CRNS sensors which we use to simulate the measurement of neutron count rates, but not to retrieve a soil moisture product in the context of this study. The focus of this study is to provide a consistent framework to simulate virtual observations which could be used to experiment with soil moisture retrieval (by using each technique for itself or by combining their strengths). Including this analysis step, however, is beyond the scope of this study. In the revised version of the manuscript, this will again be emphasized in the introduction as well as in the conclusions section.

**RC:** *The use of microwave observations (e.g. from SAR data) would have been more appropriate. I suggest that the authors restructure the text to make it more balanced. Some parts can simply be put in the supplementary material. The remote sensing case study is very weak.*

AR: We agree that SAR is an important approach to derive soil moisture via remote sensing. As explained in the previous reply, we acknowledged this in the introduction, but focused only on optical remote sensing for the examples. We will make this point clearer in the revised version.

Despite the strengths of SAR, we think that optical remote sensing is likewise worth to be analyzed in this context. Especially in the face of the limited penetration depth of active microwave sensors (or none at all in case of radar in dense forests with e.g. S1 C-band in dense forests), optical imagery can reveal water induced effects in vegetation vitality, thus potentially integrate over the entire root zone. Consequently, vegetation indices that show reduction in leaf water content, chlorophyll or even soil moisture directly (for sparse vegetation and bare soil) can be useful to derive strategies to manage irrigation systems, assess wildfire risk and other disturbance. They are widely applied (e.g. Li et al., 2022), also because of their usually higher spatial and temporal resolution, e.g. Buitink et al, 2020. Moreover, there are hardly any SAR-based RTMs available, which use high spatial resolution S-1 data - one exception is https://doi.org/10.3390/rs12183037,

but the model is not publicly available, via GIThub etc. For other wavelenghts like L-band, the respective spatial resolutions is approx. 1000 m, so far to coarse for detailed analyses on the spatial scale of the vJFC. . We will add these explanations to the respective RS-section.

We are not sure which "parts can simply be put in the supplementary material" and would appreciate concrete suggestions on this matter.

RC: *MAJOR COMMENT 2: While reading the paper, I wondered how the virtual truth was developed. It is only clear after carefully reading the methodology, I would suggest adding a paragraph at the end of the introduction clearly describing it. For example, I was expecting virtual experiments with time-varying soil moisture, but this is not the case. This should be mentioned, and I also wondered if this might be a strong limitation of the current design of the framework. I would suggest that the authors discuss this point.*

Again, we would like to stress that the intended focus of the manuscript is the framework, not the presented application examples. Therefore, we would like to keep the general aspects on the generation of the virtual truth generic and in the Methodology section. We do not think they should already be spelled out in the introduction, as they constitute a major part of the conceptual novelty. Therefore, we described these aspects already in Section 2.2.1" Construction: extent, resolution, recombination". The current focus on single snapshots in time is also mentioned there. However, we will improve the explanation that this is more a current pragmatic reduction, rather than a fundamental limitation of the concept.

Moreover, we will add an outline and justify the structure of the manuscript at the end of the intro section. That might help to manage expectations with regard to the content of the various sections/subsections.

RC: *MAJOR COMMENTS 3: The second experiment (sierra-neutronica) is briefly described. While potentially interesting, as it was in the current version, it is described too briefly. I would suggest either removing the experiment or improving its description and relevance. What understanding do we gain from such an experiment?*

AR: The case study "sierra_neutronica" focusses on the effect of relief, which is potentially affecting all of the three involved sensors, i.e. CRNS, optical remote sensing and hydrogravimetry. As the understanding of relief effects on CRNS-signal is especially limited, we chose to focus on this sensor in the example. We will extend this justification in the manuscript.

RC: *SPECIFIC COMMENTS L37: for remote sensing, active and passive microwave are mentioned here. In the paper optical remote sensing is considered, and also thermal data are used for retrieving soil moisture. Please improve the text here.*

AR: We do not quite understand this request. The mentioned section in the introduction introduces the relevant state-of-the-art methods for measuring soil moisture beyond the point scale. It does not and should not describe the methods applied in the three case studies. Therefore, we suggest to leave this section as it is.

RC: *L39: Also, gamma-ray technique is worth to be mentioned here.*

AR: Thanks for the suggestions, will be added.

RC: *L46: "(reference welcome)" something is missing here*

AR: Will be removed.

RC: *L52: Just a comment, interesting to see that such "virtual campaign" is similar to the concept of "digital twin" for developing scenario on the potential behaviour of the Earth System. The connection between the two concepts might be mentioned here.*

AR: Thanks for this suggestion. Indeed, we had discussed this point already during the writing of the manuscript. However, we decided against it as the vJFC's primary use is the provision of a virtual testbed for different sensors. These testbeds *may* resemble real systems, but (as in two of our three examples) can also be completely synthetic. Thus, we found the analogy to "digital twin" misleading.

RC: *L72: The free availability of scripts and data is very welcome, but it cannot be considered as a requirement for developing a virtual landscape.*

AR: We agree that this is not a strict scientific requirement. However, for practical reasons we consider it a very beneficial and desirable point, especially in the face of re-use and re-combination of the existing building blocks. We will modify the sentence accordingly.

RC: *L118: The spatial scale of remote sensing "$10^-1\ldots10^2$ m" is not correct; it should be, at least, "$10^1\ldots10^3$ m", if high-resolution data for soil moisture and biomass are considered.*

AR: The given figure refers to the *support* (see Blöschl and Sivapalan, 1995) of the single sensor unit, i.e. the spatial extent that influences the single measurement. For remote sensing, this translates to the area covered by a single raster cell. Starting from UAV imagery with ground resolution at the cm-scale to satellite products with several hundred meters resolution. However, as some related remote sensing products also feature km-sizes grid cells, we will correct the given range to "$10^-1\ldots10^3$".

RC: *L124: "single fixed point in time". This is mentioned here for the first time, likely better to underline it before (see also the second major comment).*

AR: This is not a fundamental restriction of the vJFC, but just a pragmatic decision for its current implementation. See also reply to MAJOR COMMENT 2.

RC: *L138: The "pattern" concept is not clear here. I would suggest clarifying.*

AR: 'pattern' is a meta-component defining spatial patterns, which can be referred to in the construction of other compartments to create coherent spatial patterns. For example, a pattern defining the extent of grassland and forest may serve as a basis for generating coherent compartments of vegetation ("grass" and "forest"), soil density ("medium" and "low") and soil moisture ("medium" and "low") corresponding to these areal entities.

We will add this explanation to the respective section.

RC: *L146-151: Also here, the different combination techniques are not fully clear. I would suggest clarifying.*

AR: The combination techniques describe the options how the different compartments are intersected by mixing or replacement.

We will extend the description to the respective section.

RC: *L281: I would add "spatial", i.e., "its spatial variability".*

AR: The respective subheading is "A landscape with maximized field-scale heterogeneity". We think that "field-scale" clearly implies that this refers to *spatial* variability. We also think that "field scale" is more informative than just "spatial", as we address the variability at the scale of several meters to decameters, opposed to e.g. cm-scale variability in soil moisture (which can also be relevant, but which we do not cover).

RC: *Table 2: The combination for soil moisture should be 12 (4x3), not 8. Why?*

AR: The number of 8 combinations results from the fact that the two values for mean soil moisture state "dry" and "wet" can only be combined with the profile "homogenous". In other words, a soil profile with a dry topsoil

cannot have a decreasing profile shape (as it would result in negative soil moisture in greater depths). If it has a increasing profile shape, its mean soil moisture is 33%.

**RC:**  *Figure 3: In the caption, "top" and "bottom" should be "left" and "right". "tops layer" should be "top layer".*

 AR:  Agreed, will be fixed.

**RC:**  *L367-368: Indeed, no soil moisture products from optical data is currently being developed.*

 AR:  We assume the line numbering is a mistake, as line 367 contains the subheading "3.2 sierra_neutronica: Synthetic mountains to explore topographic effects". Concerning the productions of a soil moisture product, please see reply to MAJOR COMMENT 1.

**RC:**  *Figure 6: It is missing the y-label in the plot on the right.*

 AR:  Will be added.

**References**

Buitink, J., A. M. Swank, M. van der Ploeg, N. E. Smith, H.-J. F. Benninga, F. van der Bolt, C. D. U. Carranza, G. Koren, R. van der Velde, und A. J. Teuling (2020): „Anatomy of the 2018 agricultural drought in the Netherlands using in situ soil moisture and satellite vegetation indices". Hydrology and Earth System Sciences 24, Nr. 12: 6021–31. https://doi.org/10.5194/hess-24-6021-2020.

Blöschl, G., und M. Sivapalan (1995): „Scale Issues in Hydrological Modelling: A Review". Hydrological Processes 9, Nr. 3–4 (April 1995): 251–90. https://doi.org/10.1002/hyp.3360090305.

Larson, K. M., J. J. Braun, E. E. Small, V. U. Zavorotny, E. D. Gutmann and A. L. Bilich (2010): GPS Multipath and Its Relation to Near-Surface Soil Moisture Content, IEEE Journal of Selected Topics in Applied Earth Observations and Remote Sensing, 3(1), 91-99, https://doi.org/10.1109/JSTARS.2009.2033612.

Li, W., Migliavacca, M., Forkel, M. et al. (2022): Widespread increasing vegetation sensitivity to soil moisture. Nat Commun 13, 3959 . https://doi.org/10.1038/s41467-022-31667-9

---

## Author Comment (AC2)

**Author Response to Reviewer 2**

**Virtual joint field campaign: a framework of synthetic landscapes to assess multiscale measurement methods of water storage**

Till Francke et al.

*Geoscientific Model Development Discuss.*, `doi:10.5194/gmd-2024-106`

**RC:** *Referee Comment*,    AR: *Author Response*,    ☐ Manuscript text

Dear Reviewer 2,

thank you very much for your positive response, and for the time and effort spent to examine the manuscript and the data set.

Your comments are very helpful in improving the quality of our manuscript. Please find a point-by-point reply below on how we intend to implement them. If you feel that some of these changes would not satisfy the needs you indicated, we would appreciate further advice on these matters.

Kind regards,
Till Francke (on behalf of the author team)

**Comments and responses**

**RC:** *[...] The paper is well written and structured but there are a few issues that need addressing prior to publication – I don't believe any are particularly difficult to address.*

 AR: We are grateful for the positive evaluation of the manuscript and the effort invested by both reviewers.

**RC:** *The abstract would benefit from a sentence setting out the application context (e.g. soli moisture sensing). This is very nicely explained the introduction but the abstract starts on, for me, a rather technical note.*

 AR: We will modify the abstract to better reflect this aspect as such (changes in *italics*):

> The major challenge of measurement methods *of environmental variables* beyond the point scale is their complex interpretation in the light of landscape heterogeneity. For example, methods like cosmic-ray neutron sensing, remote sensing, or hydrogravimetry are all able to provide an integral value on the water storage, representative for their individual measurement volume. [...] The present study demonstrates virtual observations *of water storage* with Cosmic Ray Neutron Sensing, Hydrogravimetry, and Remote Sensing in three exemplary landscapes.

**RC:** *I appreciate the idea of the vJFC is to be quite generic (sentences around line 100), but I wonder if the title of the framework could be more specific to water storage applications? I think this would more accurately reflect what the code does at this stage – happy for you to argue against this if other applications are int eh pipeline.*

**AR:** Indeed, although the idea of the vJFC is quite generic, the current manuscript clearly focuses on applications for water storage. However, already the included examples also allow considering other target variables as well: both CRNS and remote sensing observations can also be used to infer biomass. We have indicated this in the Methods and also in the Conclusion. Likewise, if e.g. virtual temperature observations were included, the estimation of evaporation might also be envisioned. As we have not further elaborated these ideas so far, we would like to stick to just briefly mentioning this option, but still keep the name of the framework sufficiently generic.

**RC:** *My main concern about the paper is that there is not overarching explanation or justification for the choice of case studies. Some are more detailed than others and the conclusions are essentially written without references to the case studies. I appreciate the test cases are simply illustrations of a huge number of potential applications, but a rational should be given for the choices, especially as some but not all of the measurement types are presented for each test case. How do the test cases come together to demonstrate the codebase effectively? And on that point do you do any tests (e.g. unit tests) to check the code is working as expected?*

**AR:** A similar point was also critizied by reviewer 1, so we would like to repeat the reply here:

Our focus was on the presentation of the framework, not on specific case studies. We used one realization (hexland_tracks) to demonstrate the applicability *across different sensor types*. As such, the other two case studies merely underline the versatility of the framework *across different landscape realizations*, specifically in representing terrain or resembling non-artificial landscapes.

We will clarify our focus on the framework and the illustrative and selective nature of our case studies and support this with the following table:

Table 1: Overview realizations

| realization | description | generated virtual observations |
| --- | --- | --- |
| *hexland_tracks* | synthetic landscape with max. contrasts | CRNS, remote sensing, hydrogravimetry |
| *sierra_neutronica* | synthetic landscape with high-relief | CRNS |
| *agia* | realistic Mediterranean landscape | CRNS |

This table should help to illustrate how "test cases come together to demonstrate the codebase effectively", as phrased by Reviewer 2. Concerning tests of the code, the package includes a dedicated visualisation functions (which also served to create the illustrations in the paper). We will add a sentence to section "Scripts and external model code" to indicate this option for verification.

**RC:** *I'm also not convinced I properly understood section 2.2.2 (L142). When merging if a compartment properties are replaced how is that different to stacking. Could some form of visualization be added to help with understanding how the landscapes are built in 2.2.2?*

We will add a figure similar to the one below to illustrate the different modes of combining the compartments:

[Figure]

Figure 1: Simplified examples emonstrating the different modes of combining compartments into realizations

**RC:** *I think GMD requires version numbers in the title?*

AR: The guidelines state " the title page must include the title (concise but informative, including model name and version number *if a model description paper*)". As we see our study clearly in the category "Methods for assessments of models", we have not included a version number in the title.

**RC:** *L46: Check '(reference Welcome)'*

AR: Will be fixed.

**RC:** *L373: This case study is introduced due to its importance for all three observation types, but only CRNS is presented. Why not present results form all three instrument types? At the very least a clear justification for not doing this is needed, especially as the justification is currently broader than what's presented. If the results are not particularly interesting for some reason could they be included nevertheless in a supplement? I appreciate it's likely not necessary to go into great detail about results form every test case, they are primarily illustrative, but a rational is needed for the choices about what to present (see my main comment). To be clear, I'm not advocating for more test cases and sensor examples, but the explanation of why various virtual sensor examples have been chosen needs to be much stronger.*

AR: We appreciate the reviewer understanding the illustrative nature of the examples. In addition to our reply on

Reviewer 2's respective main comment (see above), we will strengthen the justification of the selection in the beginning of section 3.

**RC:** *L485: When 'relevant the aspects that merit further analysis' are presented these should be linked with the case study where they emerged when relevant. This basically links in with my main critique of the paper – the purpose the test cases and rational of their choice is not well summarized and then not well used to support the conclusions. The conclusions could disaggregate between perceived applications and those illustrated by case studies as a way to link the case studies and conclusions.*

**AR:** Section 4 is entitled "Conclusions and outlook". Due to the illustrative nature of the case studies and the focus of the paper on the framework rather than single aspects, we did not intend to draw substantial conclusions from (very briefly analyzed) results of the case studies. Instead, the mentioned list rather focuses on the "Outlooks", i.e. the envisioned scientific questions that can be further elaborated with the presented examples. However, we will also annotate the existing list of bullet points with abbreviations to indicate which case studies could be used to address the raised questions.

**RC:** *L510: I'd like to see the limitations around assuming no measurement error introduced before the case studies, and certainly not in the conclusions (apologies if I missed this earlier in the manuscript). I was thinking about this issue while reading the case studies.*

**AR:** It is correct, that the presented examples do not consider any measurement errors. However, this is purely a decision made for the sake of keeping the examples simple. It is, by no means, a limitation of the concept of the vJFC. On the contrary, the amount of error and its effect can be systematically analyzed with this framework. We will clarify this in the respective paragraph.

Concerning the position of this aspect in the text, we intended to start with the bullet point list as related to / derived from the case studies. After that, some more general ideas, irrespective of the case studies, are presented. We will improve the wording to make this clearer.

**RC:** *L517: Computational expense is often mentioned as a barrier, but I don't think estimates of the computational expense are ever given. This would be very useful practical information for anyone using the package. Do I need a HPC system for this or is this expensive in a single computer context? As with the measurement error limitations this needs presenting before the conclusions.*

**AR:** We agree. While the requirements for the employed radiative transfer model are relatively modest, the neutron and hydrogravimetric simulations require computational power of several single-CPU-days per realization. We will add this information to suitable section.

---

## Author Response (AR1)

**Author Response to Editor after revision**

**Virtual joint field campaign: a framework of synthetic landscapes to assess multiscale measurement methods of water storage**

Till Francke et al.
*Geoscientific Model Development Discuss.,* `doi:10.5194/gmd-2024-106`

**EC:** *Editor Comment*,     AR: *Author Response*,     ☐ Manuscript text

Dear Jeffrey Neal,

thank you very much for taking the responsibility as an editor for this submission. Given the difficulty to find an editor, we appreciate this even more.

Dear reviewers, thank you for the time and effort you have spent to examine the manuscript. Your comments provided helpful guidance, and addressing them has certainly improved the comprehensibility of the manuscript and the usability of the data set.

Please find a point-by-point reply to the reviewers' comments below. We have implemented all the changes along the our suggestions we made in the interactive discussion. For transparency, we provide a document highlighting the tracked changes and a version with the final formatting. We hope that the revised manuscript and data meet your expectations and are now suitable for publication in GMD.

Thanks again for your effort!

Kind regards,
Till Francke (on behalf of the author team)

**Reviewer1: Comments and responses**

**RC:** *The paper is well written, well-structured and clear. The topic is surely of interest for the readers of Geoscientific Model Development (GMD) as the paper introduces a new virtual framework to assess measurement methods of soil moisture and biomass in controlled experiments.*

 AR:   We appreciate this positive perception of our study.

**RC:** *MAJOR COMMENT 1: The virtual framework is, in principle, suitable for any measurement technique. The paper analyses three methods: cosmic-ray, remote sensing and gravimetric measurements. However, while cosmic-ray are well developed and described, this is not the case for remote sensing and gravimetric measurements. Remote sensing and gravimetric measurements are only used in the hexland_tracks experiment, cosmic-ray in all the experiments.*

**AR:** We agree that the framework could be suitable for various measurement techniques. We tried to demonstrate this with our case study "hexland_tracks", in which CRNS, passive optical remote sensing and hydrogravimetry are used. These examples are by no means exhaustive: active microwave remote sensing, GNSS reflectometry or ground penetrating radar are other possible examples. However, as these other methods require specific expertise, we must confine ourselves to our area of knowledge. Thus, we can only invite the respective communities to this endeavour, but cannot cover more possible examples within the case studies.

Our focus was on the presentation of the framework, not on specific case studies. We used one realization (hexland_tracks) to demonstrate the applicability *across different sensor types*. As such, the other two case studies merely underline the versatility of the framework *across different landscape realizations*, specifically in representing terrain or resembling non-artificial landscapes.

We have clarified our focus on the framework and the illustrative and selective nature of our case studies in the beginning of section 3 (ll. 284) and supported this with the following the newly-added Table 2:

Table 1: Overview of realizations presented as case studies

| realization | description | generated virtual observations | | |
| --- | --- | --- | --- | --- |
| | | CRNS | RS | HG |
| *hexland_tracks* | synthetic landscape with max. contrasts | X | X | X |
| *sierra_neutronica* | synthetic landscape with high-relief | X | - | - |
| *agia*∗ | realistic Mediterranean landscape | X | - | - |

**RC:** *In addition, optical remote sensing data are considered, but currently no soil moisture products from optical data are available, only results from scientific papers.*

**AR:** The output of the radiative transfer model is spectral reflectance (resampled to the 13 bands of Sentinel 2). It is correct that we do not use the reflectance data to retrieve a soil moisture product, but just illustrate the use of the spectral response by more common indices related to the presence of water and chlorophyll (NDV, NDWI, VWC). The same essentially applies to the virtual CRNS sensors, which we use to simulate the observation of neutron count rates, but not to retrieve a soil moisture product in the context of this study. The focus of this study is to provide a consistent framework to simulate virtual observations, which could be used to experiment with soil moisture retrieval (by using each technique for itself or by combining their strengths). Including this analysis step, however, is beyond the scope of this study. To explain this, the end of section 2.2.4 Reconstruction now reads

> they [the reconstructions] are not elaborated for the case studies presented in this paper, as it focuses on the presentation of the overall framework. Nevertheless, they are of high relevance for envisioned follow-up studies."

Additionally, the following sentence was added to the conclusion section:

> The analysis of the presented case studies focused on the respective virtual observations, a subsequent reconstruction of the target variable was not covered here, but will be of utmost interest in follow-up studies.

**RC:** *The use of microwave observations (e.g. from SAR data) would have been more appropriate. I suggest that the authors restructure the text to make it more balanced. Some parts can simply be put in the supplementary material. The remote sensing case study is very weak.*

AR: We agree that SAR is an important approach to derive soil moisture via remote sensing. As explained in the previous reply, we acknowledged this in the introduction, but focused only on optical remote sensing for the examples. We improved the justification for this in the respective RS-section:

> Despite the strengths of active microwave sensors, we started with a focus on optical remote sensing in the vJFC. Especially in the face of the limited penetration depth of active microwave sensors, optical imagery can reveal water-induced effects in vegetation vitality, thus potentially integrating over the entire root zone. Corresponding spectral indices are widely applied (e.g. Li et al., 2021),, also because of their usually higher spatial and temporal resolution, (e.g. Buitink et al., 2020).

In our manuscript we are referencing each of the included figures, because we think they help understanding the presented facts. Thus, we were not sure which "parts can simply be put in the supplementary material". Since we did not get any more specific indication to this in the interactive discussion phase, we decided to leave the figures in place.

**RC:** *MAJOR COMMENT 2: While reading the paper, I wondered how the virtual truth was developed. It is only clear after carefully reading the methodology, I would suggest adding a paragraph at the end of the introduction clearly describing it. For example, I was expecting virtual experiments with time-varying soil moisture, but this is not the case. This should be mentioned, and I also wondered if this might be a strong limitation of the current design of the framework. I would suggest that the authors discuss this point.*

AR: Again, we would like to stress that the intended focus of the manuscript is the framework, not the presented application examples. Therefore, we would like to keep the general aspects on the generation of the virtual truth generic and in the Methodology section, i.e. not naming specific aspects of the selected case studies. We think that neither the general features of the vJFC nor the specifics of the case studies belong into the introduction, as their development constitutes a major part of the conceptual novelty. Thus, we described the generic aspects in Section 2.2.1" Construction: extent, resolution, recombination". The current focus on single snapshots in time is also mentioned there:

> Due to the involved heavy computational demands for some of the virtual sensors, the vJFC does not consider any continuous temporal dimension, i.e. all realizations refer to a single fixed point in time. However, multiple snapshots over time could be represented with different realizations.

Moreover, we have extended the description of the structure of the manuscript at the end of the intro section (ll. 98). We hope this helps to manage expectations with regard to the content of the various sections/subsections.

**RC:** *MAJOR COMMENTS 3: The second experiment (sierra-neutronica) is briefly described. While potentially interesting, as it was in the current version, it is described too briefly. I would suggest either removing*

*the experiment or improving its description and relevance. What understanding do we gain from such an experiment?*

AR:   The case study "sierra_neutronica" focusses on the effect of relief, which is potentially affecting all of the three involved sensors, i.e. CRNS, optical remote sensing and hydrogravimetry. As the understanding of relief effects on CRNS-signal is especially limited, we chose to focus on this sensor in the example. We have rephrased and extended this justification in section 3.2 of the manuscript.

RC:   *SPECIFIC COMMENTS L37: for remote sensing, active and passive microwave are mentioned here. In the paper optical remote sensing is considered, and also thermal data are used for retrieving soil moisture. Please improve the text here.*

AR:   We do not quite understand this request. The mentioned section in the introduction introduces the relevant state-of-the-art methods for measuring soil moisture beyond the point scale. It does not and should not describe the methods applied in the three case studies. Therefore, we suggest to leave this section as it is.

RC:   *L39: Also, gamma-ray technique is worth to be mentioned here.*

AR:   Thanks for the suggestions, has been added.

RC:   *L46: "(reference welcome)" something is missing here*

AR:   Has been removed.

RC:   *L52: Just a comment, interesting to see that such "virtual campaign" is similar to the concept of "digital twin" for developing scenario on the potential behaviour of the Earth System. The connection between the two concepts might be mentioned here.*

AR:   Thanks for this suggestion. Indeed, we had discussed this point already during the writing of the manuscript. However, we decided against it as the vJFC's primary use is the provision of a virtual testbed for different sensors. These testbeds *may* resemble real systems, but (as in two of our three examples) can also be completely synthetic. Thus, we found the analogy to "digital twin" misleading.

RC:   *L72: The free availability of scripts and data is very welcome, but it cannot be considered as a requirement for developing a virtual landscape.*

AR:   We agree that this is not a strict scientific requirement. As we are targeting and advocating intense re-use and re-combination of the existing building blocks, and the potential cross-disciplinary analysis of the results, we have left the bullet point in place, but weakened the wording.

RC:   *L118: The spatial scale of remote sensing "$10^{-}1\ldots 10^2$ m" is not correct; it should be, at least, "$10^1\ldots 10^3$ m", if high-resolution data for soil moisture and biomass are considered.*

AR:   The given figure refers to the *support* (see Blöschl and Sivapalan, 1995) of the single sensor unit, i.e. the spatial extent that influences the single measurement. For remote sensing, this translates to the area covered by a single raster cell. Starting from UAV imagery with ground resolution at the cm-scale to satellite products up to a resolution of tens of kilometers.We have corrected the range to "$10^{-2}\ldots 10^4$ m".

RC:   *L124: "single fixed point in time". This is mentioned here for the first time, likely better to underline it before (see also the second major comment).*

AR:   This is not a fundamental restriction of the vJFC, but just a pragmatic decision for its current implementation. See also reply to MAJOR COMMENT 2.

**RC:** *L138: The "pattern" concept is not clear here. I would suggest clarifying.*

AR:  'pattern' is a meta-component defining spatial patterns, which can be referred to in the construction of other compartments to create coherent spatial patterns. For example, a pattern defining the extent of grassland and forest may serve as a basis for generating coherent compartments of vegetation ("grass" and "forest"), soil density ("medium" and "low") and soil moisture ("medium" and "low") corresponding to these areal entities.

We have added the following explanation to the respective section:

> For example, a pattern defining the extent of grassland and forest may serve as a basis for generating coherent compartments of vegetation ("grass" and "forest"), soil density ("medium" and "low") and soil moisture ("medium" and "low") corresponding to these areal entities.

**RC:** *L146-151: Also here, the different combination techniques are not fully clear. I would suggest clarifying.*

AR:  The combination techniques describe the options how the different compartments are intersected by mixing or replacement.

We have added the following figure to make this clearer:

[Figure]

Figure 1: Cases of combination if different compartments to create a realization. Grey boxes denote empty cells, i.e. "no data"

**RC:** *L281: I would add "spatial", i.e., "its spatial variability".*

AR:  The respective subheading is "A landscape with maximized field-scale heterogeneity". We think that "field-scale" clearly implies that this refers to *spatial* variability. We also think that "field scale" is more informative than just "spatial", as we address the variability at the scale of several meters to decameters, opposed to e.g. cm-scale variability in soil moisture (which can also be relevant, but which we do not cover).

**RC:** *Table 2: The combination for soil moisture should be 12 (4x3), not 8. Why?*

**AR:** The number of 8 combinations results from the fact that the two values for mean soil moisture state "dry" and "wet" can only be combined with the profile "homogenous". In other words, a soil profile with a dry topsoil cannot have a decreasing profile shape (as it would result in negative soil moisture in greater depths). If it has a increasing profile shape, its mean soil moisture is 33%. We have added a respective explanation to the footer of Table 3.

**RC:** *Figure 3: In the caption, "top" and "bottom" should be "left" and "right". "tops layer" should be "top layer".*

**AR:** Fixed.

**RC:** *L367-368: Indeed, no soil moisture products from optical data is currently being developed.*

**AR:** We assume the line numbering was a mistake, as line 367 contained the subheading "3.2 sierra_neutronica: Synthetic mountains to explore topographic effects". Concerning the productions of a soil moisture product, please see reply to MAJOR COMMENT 1.

**RC:** *Figure 6: It is missing the y-label in the plot on the right.*

**AR:** Label added.

**Reviewer2: Comments and responses**

**RC:** *[...] The paper is well written and structured but there are a few issues that need addressing prior to publication – I don't believe any are particularly difficult to address.*

**AR:** We are grateful for the positive evaluation of the manuscript and the effort invested by both reviewers.

**RC:** *The abstract would benefit from a sentence setting out the application context (e.g. soli moisture sensing). This is very nicely explained the introduction but the abstract starts on, for me, a rather technical note.*

**AR:** We have modified the abstract to better reflect this aspect.

**RC:** *I appreciate the idea of the vJFC is to be quite generic (sentences around line 100), but I wonder if the title of the framework could be more specific to water storage applications? I think this would more accurately reflect what the code does at this stage – happy for you to argue against this if other applications are int eh pipeline.*

**AR:** Indeed, although the idea of the vJFC is quite generic, the current manuscript clearly focuses on applications for water storage. However, already the included examples also allow considering other target variables as well: both CRNS and remote sensing observations can also be used to infer biomass. We have indicated this in the Methods (2.1) and also in the Conclusion. Likewise, if e.g. virtual temperature observations were included, the estimation of evaporation might also be envisioned. As we have not further elaborated these ideas so far, we would like to stick to just briefly mentioning this option, but still keep the name of the framework sufficiently generic.

**RC:** *My main concern about the paper is that there is not overarching explanation or justification for the choice of case studies. Some are more detailed than others and the conclusions are essentially written without references to the case studies. I appreciate the test cases are simply illustrations of a huge number of*

*potential applications, but a rational should be given for the choices, especially as some but not all of the measurement types are presented for each test case. How do the test cases come together to demonstrate the codebase effectively? And on that point do you do any tests (e.g. unit tests) to check the code is working as expected?*

AR: A similar point was also criticized by reviewer 1, so we would like to repeat the reply here:

Our focus was on the presentation of the framework, not on specific case studies. We used one realization (hexland_tracks) to demonstrate the applicability *across different sensor types*. As such, the other two case studies merely underline the versatility of the framework *across different landscape realizations*, specifically in representing terrain or resembling non-artificial landscapes.

We have clarified our focus on the framework and the illustrative and selective nature of our case studies in the beginning of section 3 and supported this with the the newly-added Table 2. This table should help to illustrate how "test cases come together to demonstrate the codebase effectively", as phrased by Reviewer 2. Concerning tests of the code, the package includes a dedicated visualisation functions (which also served to create the illustrations in the paper). We have added a bullet point to the end of section "Scripts and external model code" to indicate this option for verification and interactive inspection.

RC: *I'm also not convinced I properly understood section 2.2.2 (L142). When merging if a compartment properties are replaced how is that different to stacking. Could some form of visualization be added to help with understanding how the landscapes are built in 2.2.2?*

AR: We have added figure 2 to address this concern:

[Figure]

Figure 2: Cases of combination if different compartments to create a realization. Grey boxes denote empty cells, i.e. "no data"

RC: *I think GMD requires version numbers in the title?*

AR: The guidelines state " the title page must include the title (concise but informative, including model name and

version number *if a model description paper*)". As we see our study clearly in the category "Methods for assessments of models", we have not included a version number in the title.

**RC:** *L46: Check '(reference Welcome)'*

AR: Fixed.

**RC:** *L373: This case study is introduced due to its importance for all three observation types, but only CRNS is presented. Why not present results form all three instrument types? At the very least a clear justification for not doing this is needed, especially as the justification is currently broader than what's presented. If the results are not particularly interesting for some reason could they be included nevertheless in a supplement? I appreciate it's likely not necessary to go into great detail about results form every test case, they are primarily illustrative, but a rational is needed for the choices about what to present (see my main comment). To be clear, I'm not advocating for more test cases and sensor examples, but the explanation of why various virtual sensor examples have been chosen needs to be much stronger.*

AR: We appreciate the reviewer understanding the illustrative nature of the examples. In addition to the points mentioned in reply on Reviewer 2's respective main comment (see above), we have strengthened the justification of the selection in the beginning of section 3.

**RC:** *L485: When 'relevant the aspects that merit further analysis' are presented these should be linked with the case study where they emerged when relevant. This basically links in with my main critique of the paper – the purpose the test cases and rational of their choice is not well summarized and then not well used to support the conclusions. The conclusions could disaggregate between perceived applications and those illustrated by case studies as a way to link the case studies and conclusions.*

AR: Section 4 is entitled "Conclusions and outlook". Due to the illustrative nature of the case studies and the focus of the paper on the framework rather than single aspects, we did not intend to draw substantial conclusions from (very briefly analyzed) results of the case studies. Instead, the mentioned list rather focuses on the "Outlooks", i.e. the envisioned scientific questions that can be further elaborated with the presented examples. However, we annotated the existing list of bullet points with abbreviations to indicate which case studies could be used to address the raised questions. We also improved the structure to clarify which points are related to the case studies, and which are connected to the general concept of the framework.

**RC:** *L510: I'd like to see the limitations around assuming no measurement error introduced before the case studies, and certainly not in the conclusions (apologies if I missed this earlier in the manuscript). I was thinking about this issue while reading the case studies.*

AR: It is correct, that the presented examples do not consider any measurement errors. However, this is purely a decision made for the sake of keeping the examples simple. It is, by no means, a limitation of the concept of the vJFC. On the contrary, the amount of error and its effect can be systematically analyzed with this framework. We have rephrased the respective paragraph to clarify this:

> In real-world applications, measurement error or noise often poses severe restrictions on the usability of the sensor signal (e.g. due to short counting intervals for CRNS, atmospheric transmissivity for RS, instrument noise for HG). The presented case studies assumed perfect measurement conditions, i.e. no instrument error e.g. due to atmospheric conditions. This decision was made for the sake of keeping the examples simple. It is, by no means, a limitation of the concept of the vJFC. On the contrary, systematically assessing the limits of applicability or error-affected signals could be greatly facilitated with the vJFC.

Concerning the position of this aspect in the text, we intended to start with the bullet point list as related to / derived from the case studies. After that, some more general ideas, irrespective of the case studies, are presented. We have improved the wording to make this structure clearer.

**RC:** *L517: Computational expense is often mentioned as a barrier, but I don't think estimates of the computational expense are ever given. This would be very useful practical information for anyone using the package. Do I need a HPC system for this or is this expensive in a single computer context? As with the measurement error limitations this needs presenting before the conclusions.*

AR: We agree. While the requirements for the employed radiative transfer model are relatively modest, the neutron and hydrogravimetric simulations require computational power of several single-CPU-days per realization. We have added this information (l. 536):

> Generating some of the virtual observations (namely running the neutron and hydrogravimetric simulations), requires computation times on the order of several CPU-days per realization.

**References**

Blöschl, G., und M. Sivapalan (1995): „Scale Issues in Hydrological Modelling: A Review". Hydrological Processes 9, Nr. 3–4 (April 1995): 251–90. https://doi.org/10.1002/hyp.3360090305.

Larson, K. M., J. J. Braun, E. E. Small, V. U. Zavorotny, E. D. Gutmann and A. L. Bilich (2010): GPS Multipath and Its Relation to Near-Surface Soil Moisture Content, IEEE Journal of Selected Topics in Applied Earth Observations and Remote Sensing, 3(1), 91-99, https://doi.org/10.1109/JSTARS.2009.2033612.